# An oscillatory mechanism for multi-level storage in short-term memory

Kathleen P. Champion[1,7], Olivia Gozel [2,3,7], Benjamin S. Lankow[4,7], G. Bard Ermentrout [5✉] &
Mark S. Goldman [4,6✉]

Oscillatory activity is commonly observed during the maintenance of information in short-term memory, but its role remains unclear. Non-oscillatory models of short-term memory storage are able to encode stimulus identity through their spatial patterns of activity, but are typically limited to either an all-or-none representation of stimulus amplitude or exhibit a biologically implausible exact-tuning condition. Here we demonstrate a simple mechanism by which oscillatory input enables a circuit to generate persistent or sequential activity that encodes information not only in the spatial pattern of activity, but also in the amplitude of activity. This is accomplished through a phase-locking phenomenon that permits many different amplitudes of persistent activity to be stored without requiring exact tuning of model parameters. Altogether, this work proposes a class of models for the storage of information in working memory, a potential role for brain oscillations, and a dynamical mechanism for maintaining multi-stable neural representations.

[1] Department of Applied Mathematics, University of Washington, Seattle, WA 98195, USA. [2] Departments of Neurobiology and Statistics, University of Chicago, Chicago, IL 60637, USA. [3] Grossman Center for Quantitative Biology and Human Behavior, University of Chicago, Chicago, IL 60637, USA. [4] Center for Neuroscience, University of California, Davis, Davis, CA 95618, USA. [5] Department of Mathematics, University of Pittsburgh, Pittsburgh, PA 15213, USA. [6] Department of Neurobiology, Physiology, and Behavior, and Department of Ophthalmology and Vision Science, University of California, Davis, Davis, CA 95618, USA. [7] These authors contributed equally: Kathleen P. Champion, Olivia Gozel, Benjamin S. Lankow. ✉email: bard@pitt.edu; msgoldman@ucdavis.edu

The maintenance of information in short-term memory is a key component of a wide array of cognitive[1,2] and non-cognitive[3,4] functions. However, the biophysical mechanisms that enable memory storage over the seconds-long time scale remain unclear. Single-unit studies have demonstrated a neural correlate of memory maintenance in the persistent activation of neurons whose population activity spans the memory period (reviewed in refs. [2,4–6]). Theoretical studies have shown how such persistent activity can be generated by recurrent network feedback[7–9], but simple instantiations of this idea are either implausibly sensitive to mistuning or can only maintain a single elevated firing rate that is unrealistically high (the "low firing rate problem", reviewed in refs. [2,10]), limiting storage about a given item to a single bit ("on" or "off") of information.

Separately, previous studies have identified distinct bands of oscillatory activity in field potential recordings and EEG during the maintenance of working memory (reviewed in ref. [11]). Such activity can be generated through cell-intrinsic mechanisms, local circuitry, or long-range interactions[12–14]. However, it remains an open question whether oscillatory activity is necessary, sufficient, or even beneficial for working memory storage. Previous work has proposed how oscillations can contribute to a variety of memory functions such as the generation or maintenance of persistent activity[15,16]; the structuring of spatial codes through frequency coupling[17]; and the coordination, control, and gating of memory-related activity[18–30]. By contrast, other studies have suggested that oscillations could be an epiphenomenon of other computational or network mechanisms[10,31]. Here, we demonstrate a potential mechanistic role for oscillations, regardless of source or frequency, by showing how the addition of oscillatory inputs to simple recurrent feedback circuits can enable both low firing rate persistent activity and a discretely graded set of persistent firing rates that increases the information capacity of a memory network.

## Results and discussion

To illustrate the core challenges that arise when generating biologically plausible models of persistent activity, consider an idealized circuit consisting of a memory neuron (or lumped population) connected to itself through positive feedback (Fig. 1a); this basic motif of recurrent excitation is the key component of most circuit models of persistent neural activity (reviewed in ref. [2]). This simple circuit receives a brief stimulus (Fig. 1a, c, e, external input) and needs to store it through persistent activity. Stable persistent activity ($\frac{dr}{dt} = 0$) in the absence of external input is achieved only when the intrinsic decay of the neuron (represented by the term $-r$) and the recurrent drive to the neuron ($f(r)$) are equal in magnitude and cancel each other. This condition imposes two separate, but related, problems that depend on whether the firing rate function $f(r)$ is linear or nonlinear. In the typical nonlinear case (Fig. 1b, c), if the stimulus is too weak, the memory neuron's low initial firing rate provides insufficient recurrent feedback to overcome the post-stimulus intrinsic decay of activity (Fig. 1b, left of open circle). As a result, the firing rate of the network returns to a low (or zero) baseline firing rate (Fig. 1c, orange, purple, green traces). By contrast, if the stimulus is stronger, the memory neuron's initial firing provides recurrent feedback that exceeds the rate of intrinsic decay (Fig. 1b, right of open circle), leading to a reverberatory amplification of activity in which the rate rises until some saturation process brings the rate to rest at an elevated persistent level (Fig. 1c, blue and red traces). Thus, the only possibilities are that activity decays to its baseline level or that activity runs away to saturation at a high level that, for typical neuronal nonlinearities, is unrealistically large. A different problem emerges in the case in

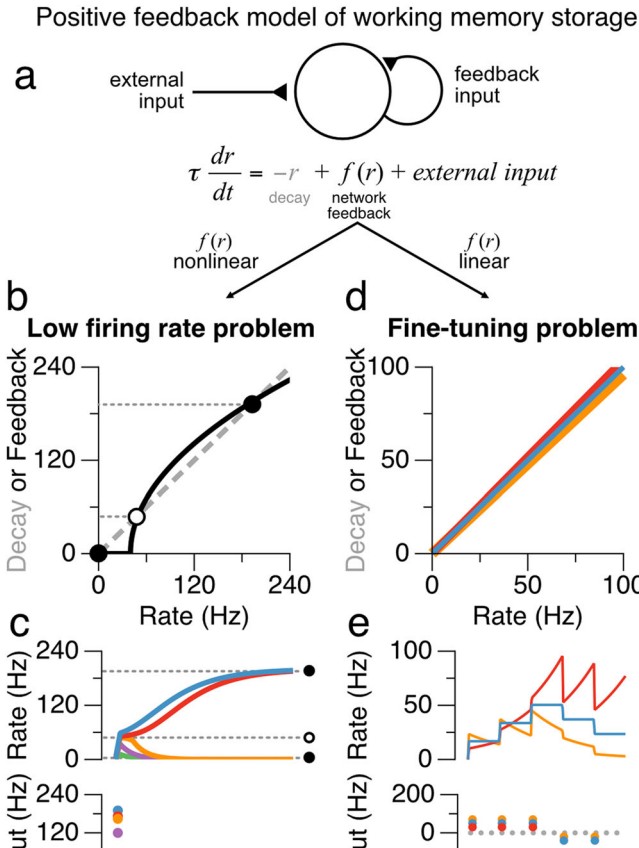

Positive feedback model of working memory storage

**a**

$$\tau \frac{dr}{dt} = \underbrace{-r}_{\text{decay}} + \underbrace{f(r)}_{\substack{\text{network} \\ \text{feedback}}} + external\ input$$

**Fig. 1 Failures of traditional positive feedback models of working memory storage. a** Simplified model illustrating key features of positive feedback models. In the absence of external input (external input = 0), changes in the firing activity $r(t)$ of a population are determined by the relative balance of network feedback (black, $f(r)$) and neuronal decay processes (gray, $-r$). **b, c** Nonlinear models typically exhibit a "low firing rate problem". **b** During the memory period when external input is absent, the intersections of the decay (gray) and network feedback (black) functions are such that there are no stable fixed points (solid circles) within the range of firing rates typically observed during persistent neural activity. **c** Firing rates below the unstable fixed point (**b, c**, open circle) decay to zero (green, purple, orange lines), whereas firing rates above the unstable fixed point run off to unrealistically high rates (red, blue lines). **d, e** Linear models exhibit the "fine tuning problem": minute changes in the strength of feedback (red: +5%, orange: −5%) relative to the tuned value (blue) result in unstable growth (red) or decay (orange).

which the firing rate function $f(r)$ is linear (Fig. 1d, e). The linearity of the rate function in this case allows a continuum of persistent rates, corresponding to the continuous set of points at which the feedback and decay lines overlap, to be stored (Fig. 1d, blue line), unlike the nonlinear case. This comes at the cost of a "fine-tuning" condition: the strength of the recurrent synapse(s) must be exactly tuned to counterbalance the strength of the rate decay; an arbitrarily small violation of this condition causes the rate to exhibit runaway growth (Fig. 1e, red trace) or decay to a low baseline (Fig. 1e, orange trace). Although presented here for a very simple example, these problems are also commonly observed in larger neural networks[32].

We next illustrate what happens when a network with the same positive feedback architecture is provided with a subthreshold

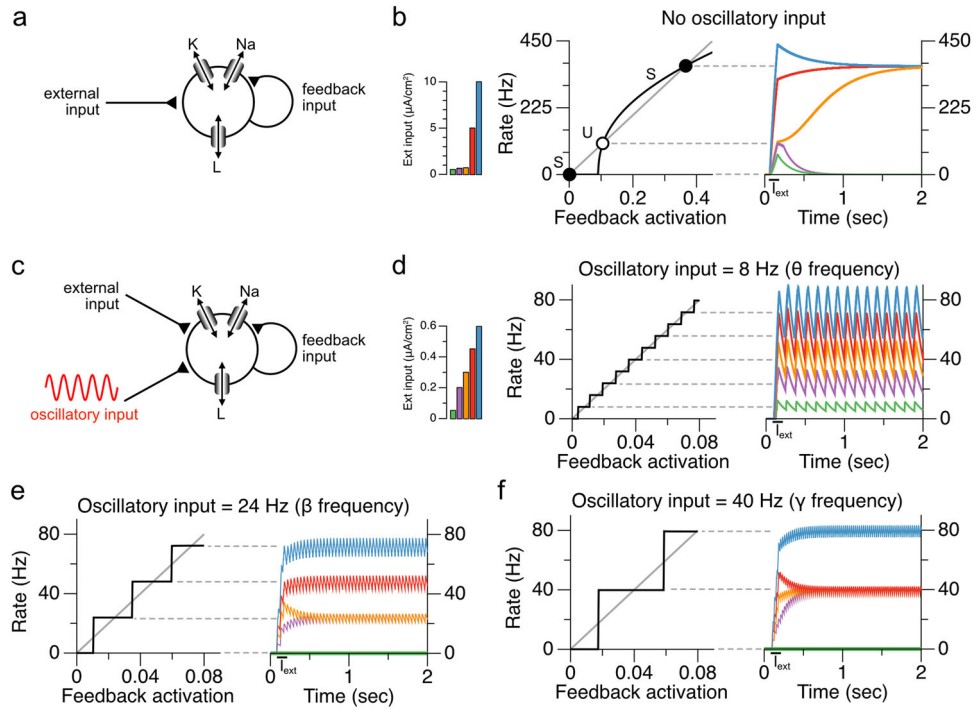

**Fig. 2 Oscillatory input allows robust maintenance of discretely graded persistent activity in a conductance-based, spiking neuron model. a** Schematic of conductance-based autapse model. The model is composed of potassium, sodium, and leak conductances, and receives feedback input ($I_{syn}$ in Eq. (1), "Methods") as well as input current from an external source. **b** Manifestation of the "low-firing-rate" problem in the conductance-based model without oscillatory input. Similar to the nonlinear firing rate model depicted in Fig. 1b, the conductance-based spiking model exhibits stable fixed points only at zero and high firing rates (filled circles). Spiking rates between these two fixed points decay to zero from below the unstable fixed point (open circle) or run off to high rates from above the unstable fixed point. Feedback activation refers to the synaptic feedback activation variable $s$ (Eq. (3), "Methods").
**c** Schematic of conductance-based neuron with the addition of an oscillatory input. **d–f** Maintenance of discretely graded persistent activity levels enabled by oscillatory input. Phase-locking to the oscillatory input creates stable fixed points at integer multiples of the oscillation frequency. There is a trade-off between the number of firing rates that can be maintained and the robustness of these fixed points, which is related to the spacing between the fixed points. **d** Lower frequency oscillations enable a larger number of closely spaced fixed points. **e, f** Higher frequency oscillations lead to fewer, more robust, fixed points. Time-varying firing rates in (**b, d–f**) are computed by smoothing the spike trains using an exponential filter with time constant equal to the model's synaptic time constant (150 ms).

oscillatory drive (Fig. 2). We demonstrate this in the more biologically realistic case of a conductance-based spiking neuron model (the Wang-Buzsaki (W-B) model of ref. [33]), which facilitates the illustration of the phase-locking phenomenon that we will describe. Without an oscillatory input, the model exhibits the "low firing-rate problem" (Fig. 2a, b) and can only maintain persistent activity at a high spiking rate or not spike at all. When a subthreshold oscillation is added to the model (Fig. 2c–f), the oscillatory drive has two effects. First, it provides extra input that allows small transient inputs to trigger low-rate spiking. Second, spiking of the memory neuron does not lead to runaway feedback because, before the feedback can run away, the oscillatory drive returns toward its trough, causing a cessation of spiking. The net result is that the spike-driven feedback becomes discretized, forming a staircase whose step heights correspond to the number of spikes emitted by the neuron per oscillation cycle (Fig. 2d–f). The phase locking of the spiking to the subthreshold oscillatory drive constrains these spike numbers to be integer multiples of the oscillation.

There are several key requirements for this mechanism to enable discretely graded persistent activity. First, the oscillation must be sufficiently strong. For very small oscillatory input amplitudes, the system resembles the case with no oscillatory input of Fig. 2b, in which external input is either too weak to cause sustained spiking so that activity returns quickly to the lower, no-spiking stable fixed point, or is strong enough to trigger spiking but then runs off to the very high upper fixed point. To

maintain discretely graded persistent activity in the recurrently connected network, the oscillation must be high enough at its peak to annihilate the no-spiking fixed point and cause spiking, and low enough at its trough to annihilate the upper fixed point and terminate spiking in each cycle.

Second, there must be some process that enables the activity from one cycle of the oscillation to carry through to the start of the next cycle and consequently enable renewed spiking as the oscillatory input heads toward its peak. For the simple case illustrated here, where all neurons receive oscillatory inputs that are perfectly aligned in phase, the mechanism enabling inter-cycle memory is a slow NMDA-like (or local dendritic) synaptic time constant[3,10,34]. Alternatively, we show in Supplementary Fig. 1 that, for a network of many neurons that receive oscillatory input with heterogeneous phases and therefore fire at staggered times, the synaptic time constant can be smaller and the time between cycles may be bridged by the firing of other neurons in the network.

Third, the single neuron model must be sufficiently nonlinear to enable phase locking to the external oscillation (Fig. 3). The key feature of this nonlinearity is that it must keep small changes in synaptic input, for example due to small changes in synaptic weights, from causing corresponding changes in firing rate. This occurs in the Wang-Buzsaki model (Fig. 3a) because small perturbations (Fig. 3b, + or − pulse) that cause transient phase shifts during the supra-threshold portion of the oscillatory cycle quickly decay away during the sub-threshold portion of the cycle

## W-B model with autapse

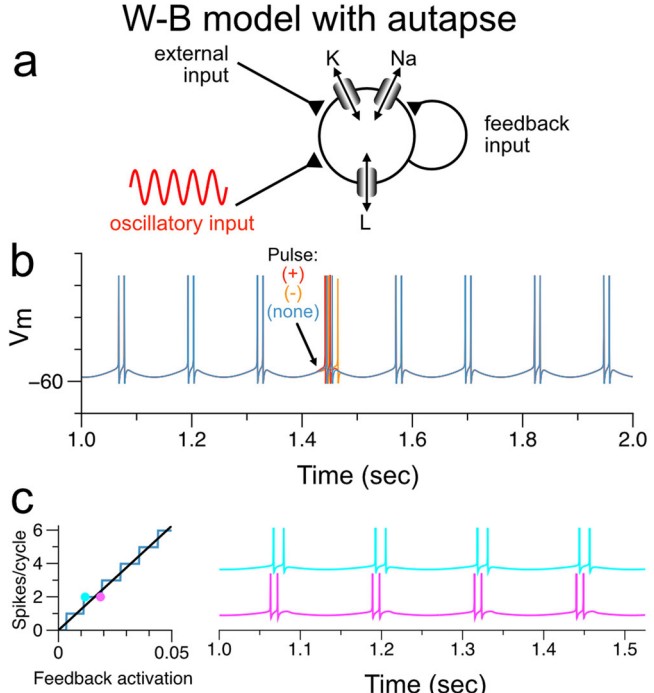

**Fig. 3 Phase-locking provides a correction to small external perturbations. a** Model schematic as in Fig. 2c. **b** A small input perturbation (+ or − pulse) causes a transient phase shift (red, phase advance; orange, phase delay) during the supra-threshold portion of the oscillatory cycle that is subsequently reset during the sub-threshold portion of the cycle. Simulation shown for the conductance-based autapse model depicted in (**a**). **c** Simulations with synaptic feedback activation held constant at two different levels (cyan and magenta points). For each synaptic feedback level, the corresponding voltage traces have the same number of spikes per cycle.

(Fig. 3b). Due to this decay, the number of spikes per cycle, and thus the output firing rate, can remain constant even when the level of sustained synaptic input changes (Fig. 3c), leading to the observed staircase in the firing rate versus feedback relation (Fig. 3c).

To understand more quantitatively the origin of this nonlinear phase-locking and the resultant staircase of firing rate versus feedback, we considered a simple integrate-and-fire model neuron with oscillatory input. For a non-leaky integrate-and-fire neuron (Fig. 4a), which has perfectly linear, non-decaying sub-threshold voltage dynamics, applying a brief pulse of external input led to a phase shift in spiking that persisted across cycles (Fig. 4b). Due to this lack of phase-resetting, when the level of sustained synaptic input is even slightly increased, it causes a progressive phase advance of spiking that builds up across cycles until an additional spike is added (Fig. 4c, compare magenta and cyan traces), increasing the firing rate slightly (note that, because the second spike occurs late in the cycle, only a single spike can fit in the next cycle). For a leaky integrate-and-fire neuron (Fig. 4d), by contrast, a brief pulse of external input leads to a transient shift in the times of the next spikes, but this phase shift decays away exponentially with the leak time constant during the subthreshold portion of the cycle (Fig. 4e; mathematical derivation shown in Supplementary Note 1). As a result, small changes in synaptic input do not build up across cycles and quite different synaptic inputs can lead to the same number of spikes each cycle (Fig. 4f).

Some degree of tuning of the feedback is required to have multiple levels of persistent activity. However, the width of the steps of the staircase provides a moderate level of robustness to

mistuning, especially for higher oscillation frequencies (Fig. 2d–f). Mechanistically, this robustness occurs because errors in the tuning of feedback that are insufficient to systematically add or subtract an extra spike per cycle do not persist from cycle to cycle, unlike in models that do not obey the three conditions described above. We illustrate this robustness to weight changes in Fig. 5, where we compare the oscillatory autapse memory model of Fig. 2d (Fig. 5a, b) to an approximately linear autapse model (see Supplementary Note 1 and refs. [35,36]) that can produce (nearly) graded persistent activity in the absence of oscillatory input (Fig. 5c, d). Each model receives an arbitrary sequence of positive and negative input pulses and must temporally accumulate and store the pulses in persistent activity. The linear spiking autapse model requires fine tuning to maintain persistent activity: very small deviations from the tuned autapse weight lead to activity that grows to a saturating level or decays to zero activity (Fig. 5c, d). In contrast, the same synaptic weight deviations have negligible effect on the accumulation and multi-level storage capability of the nonlinear spiking neuron with oscillatory drive (Fig. 5a, b).

The above examples demonstrate the basic mechanism by which oscillatory input may permit discretely graded levels of firing rate to be robustly stored in a recurrent excitatory network model of persistent activity. We next explored applications of this basic principle in the case of three different network architectures: a spatially uniform (all-to-all) network that temporally integrates its inputs (Fig. 6); a "ring-like" architecture whose activity can store both a spatial location and discretely graded levels at that location (Fig. 7); and a chain-like architecture that can generate sequences of activity with multiple discretely graded amplitudes (Fig. 8).

We first extended the demonstration of temporal integration, shown in Fig. 5, to a spatially homogeneous (all-to-all) network composed of 1000 neurons (Fig. 6a, b). This permitted us to not only examine the systematic mistuning of weights shown in the autapse network, which produces identical results to the averaged activity of the 1000 neuron network ("Methods"), but also to examine the robustness to four different sources of noise and variability (Supplementary Note 2): input noise, in which each neuron in the network received independent exponentially filtered noise added to the subthreshold oscillatory drive (Fig. 6c); noise in the connection weights, in which each synaptic weight value in the network was initialized with added random noise (Fig. 6d) or in which there was sparsity of connection weights (Supplementary Fig. 2); randomly shuffled phases of the subthreshold oscillatory drive, in which each neuron received an oscillatory signal whose phase was randomly picked from a uniform distribution at initialization (Supplementary Fig. 3a); and noisy oscillation frequency and amplitude, in which the parameters of the subthreshold oscillatory drive underwent noisy drift (generated by an Ornstein-Uhlenbeck process) during the simulations (Supplementary Fig. 3b). In all of these cases, the network was able to accurately maintain multi-level persistent activity despite moderate perturbations. Figure 6c, d and Supplementary Fig. 3b illustrate the conditions for which the magnitude of the perturbations notably affected network performance—for noise substantially less than this amount, persistent activity was well maintained over a timescale of seconds, whereas larger noise levels led to progressively larger drifts of activity. We also considered the case of perfectly correlated frequency noise across neurons, with the amplitude of oscillation-frequency noise the same as the oscillation-frequency noise in Supplementary Fig. 3b (Eq. (S.47) of Supplementary Note 2). In this case, due to not being able to average out noise across neurons, performance was worse, but for typical noise instantiations, the network still approximately maintained its level of persistent activity for multiple seconds.

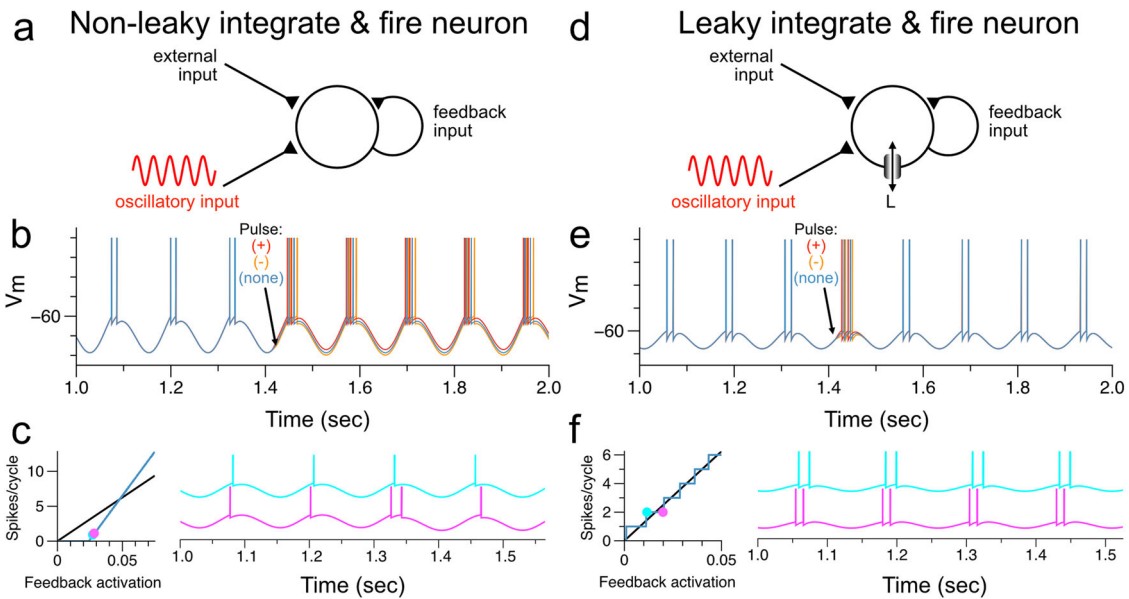

**Fig. 4 Discretely graded spiking per oscillatory cycle is related to a phase-locking mechanism enabled by a restorative decay in the membrane potential. a** Model schematic of non-leaky integrate and fire neuron receiving an oscillatory input and a feedback self-connection. **b** A perturbation (current) pulse is applied to the neuron while firing at a steady-state value of two spikes per cycle. As there is no decay in the membrane potential, a "voltage memory" is carried over from cycle to cycle, and the spiking does not phase-lock to the oscillatory drive. Synaptic feedback input is held constant in this simulation as, due to lack of phase locking, not doing so leads to activity decaying to zero or running away. **c** Non-leaky integrate and fire neuron receiving an oscillatory input and a synaptic feedback input held constant exhibits continuously graded spiking activity. For any two very close-together levels of synaptic feedback activation (cyan and magenta points), the voltage traces exhibit a different number of spikes per cycle. **d** Model schematic of leaky integrate and fire neuron receiving an oscillatory input and a feedback self-connection. **e** A perturbation (current) pulse is applied to the neuron while firing at a steady-state value of two spikes per cycle. The membrane leak in this case allows the membrane voltage to return to the same value between each period of the oscillatory cycle, essentially providing the "reset" required for phase-locking. For comparison to (**b**), the synaptic feedback input was held constant in this simulation, but, due to phase-locking, not doing so (as in Fig. 3b) leads to near-identical results. **f** Discretely graded spiking activity of a leaky integrate and fire neuron receiving an oscillatory input and a synaptic feedback input held constant. For two less close-together levels (cyan and magenta points), the voltage traces exhibit an equal number of spikes per cycle of the oscillatory input.

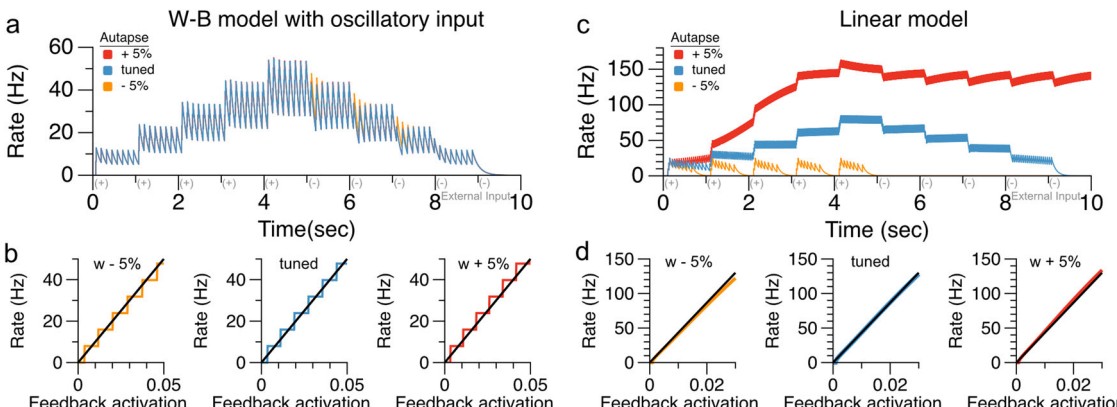

**Fig. 5 Oscillation-based integrator model exhibits more robustness to changes in the recurrent feedback weight than a traditional non-oscillation-based model. a** Responses of oscillation-based model to a sequence of positive and negative input pulses. Red and yellow traces show conditions in which the recurrent feedback strength has been detuned by ±5%, respectively. The activity levels remain persistent following detuning. **b** Steady-state firing rates as a function of synaptic feedback activation that is held at steady values; mistuning the autapse weight value by ±5% has no effect on the existence and location of the stable fixed points (intersections of black lines and horizontal stairs). **c** Responses of a traditional, approximately linear, conductance-based model of persistent neural activity (adapted from model of ref. [36]). Detuning the recurrent feedback strength by 5% (orange and red traces) causes spiking activity to decay to 0 (orange, decreased feedback strength) or run off to high rates (red, increased feedback strength). **d** Small weight changes cause systematic loss of fixed points in the traditional model.

Next we demonstrate that a similar temporal integration of inputs can also occur in spatially structured networks. We consider a classic ring model architecture commonly used to model spatial working memory tasks in which stimuli can be presented at any of various locations arranged in a circular (ring-like) layout. The model consists of a ring of neurons with local excitatory connectivity and functionally wider inhibitory connectivity (Fig. 7a, "Methods"). Such models can generate persistent activity at any spatial location along the ring, but typically have only a binary (on-off) representation at a given spatial location (Fig. 7b).

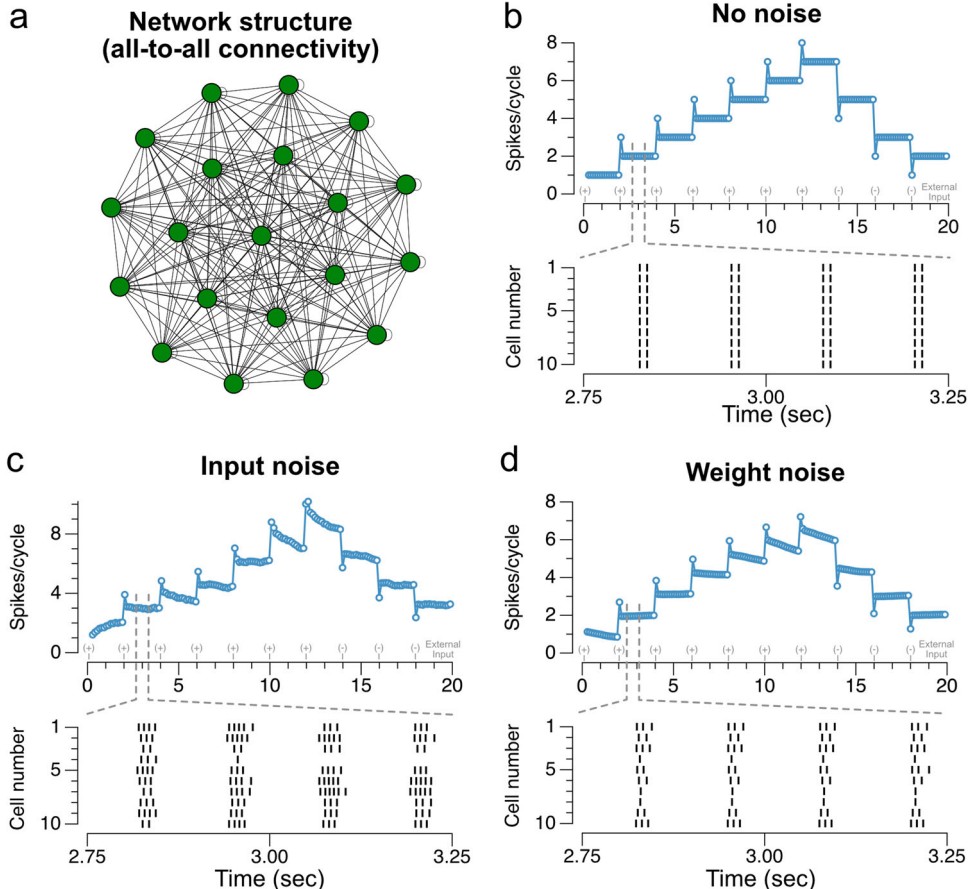

**Fig. 6 Maintenance of persistent activity and robustness to noise in a fully connected network of 1000 W-B neurons. a** Schematic of network. All units make synapses on all other units, with uniform synaptic weights (plus noise when present). **b** Spiking responses of network neurons to a sequence of positive and negative input pulses, with spike rasters plotted for the time window indicated by the dashed gray lines (random sample of 10 neurons from the 1000-neuron network). **c** Spiking responses of the network to the same input sequence in the presence of continuous external input noise. The noise had zero mean and standard deviation roughly one third the magnitude of the individual input pulses ($\sigma = 0.1\,\mu A\,ms^{0.5}/cm^2$), the point at which network activity noticeably began degrading. The network is able to maintain persistent activity despite the noisy input. **d** Spiking responses of the network initialized with random noise in the connection weights. Noise with mean zero and standard deviation of 10 times the mean connectivity strength ($\sigma = 0.055\,\mu A/cm^2$), the point at which network activity noticeably began degrading, is added to the individual connection weights between neurons. Although individual neurons in the network respond with different rates, the network is able to maintain persistent activity at many levels.

When we added an oscillatory input stimulus to such a ring model, the network could store multiple, discretely graded levels of activity at any spatial location (Fig. 7c) and could temporally integrate location-specific inputs into discretely graded levels (Fig. 7d). While the spatial memory (bump attractor) networks proposed in refs. [37–39] are capable of generating graded persistent activity, the network presented here represents, to our knowledge, the first spatial memory network to encode multi-level activity without requiring an exact tuning condition.

Recent studies have shown that memory activity during a delay period also may take the form of a sequence of activity that spans the delay[40–42]. Models of such activity typically generate chain-like patterns of activity that attain only a single, stereotyped level of firing rate. Consistent with this, when we constructed a network with a chain-like architecture (Fig. 8a), we found that, in the absence of oscillatory input, the sequential network activity either quickly decayed when the initial stimulus amplitude was too small or converged to a single saturated level of activity for larger stimuli (Fig. 8b). By contrast, in the presence of a subthreshold oscillatory input, the network could exhibit sequential activity with discretely graded amplitudes (Fig. 8c). Thus, as in the persistently active networks, the oscillatory sequential memory network could encode multiple discretized stimulus levels.

A key experimental prediction of the model is that there should be phase-locking of neural activity to oscillations, with the number of spikes per oscillation cycle increasing in a discretized manner. To test this prediction, population-wide cellular-resolution recordings in a parametric working memory task are likely necessary given noise in single-cell activity (e.g., Fig. 6c), as well as nonstationarities. EEG recordings have identified parametric variation in power at certain frequencies that encodes the graded, parametrically varied amplitude of a stimulus[43]. However, the resolution of such recordings is insufficient to determine whether representations are discretized. A larger challenge for neural recording studies of analog working memory tasks is that the set of neurons that generate such working memory, as opposed to reading it out, has not been identified in classic graded working memory tasks[44]—a readout mechanism could nonlinearly transform the representation, or pool across multiple discretized representations to obtain higher resolution readouts. The advent of high-resolution, population-wide recordings will hopefully make possible future experimental studies that can test the theoretical principle proposed in this work.

In summary, this work demonstrates a simple mechanism by which oscillatory input to a memory network can transform it

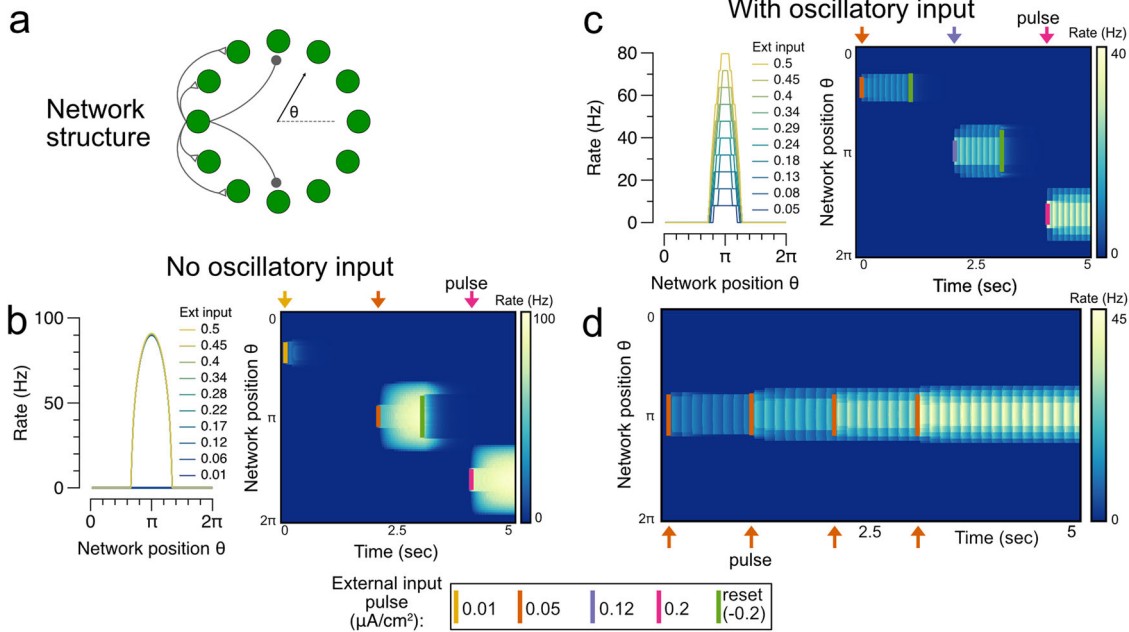

**Fig. 7 Maintenance of discretely graded bumps of persistent activity in a ring network of W-B neurons. a** Schematic of network structure. The spatial positions of neurons in the network are indexed by the angle theta from an arbitrary reference neuron. **b** Illustration of low-rate problem in a ring network of conductance-based spiking neurons without oscillatory input. Steady-state firing rate response of neurons in the network to input pulses of different amplitudes at locations centered around network position= $\pi$, and heatmaps illustrating the network's temporal firing-rate responses to short pulses of inputs at network locations labeled by colored bars. The network is unable to maintain bump activity levels between the low and high fixed points. **c** Ring network with oscillatory input is able to maintain discretely graded bumps of persistent activity. **d** Temporal integration in the ring network. Short (100 ms) input pulses to the network are temporally integrated and stored in persistent activity. For simplicity, the persistent activity in (**b**, **c**) is terminated using a pulse of inhibitory input; we do not model the origin of such termination signals.

from storing only binary amplitudes to maintaining discretely graded amplitudes of persistent activity. Memory networks using this mechanism require a cellular, synaptic, or network process that can span the period of the oscillation. The mechanism can operate at any of the many oscillation frequencies suggested to correlate with working memory storage (Fig. 2d–f): higher frequency oscillations do not require long timescale processes to span the oscillation cycle and are more robust to noise, but due to their short period may only store one or a few values; lower frequency oscillations could store more items, but require a cellular or network process with longer timescale to bridge the troughs occurring in each cycle and are more easily perturbed by noise. This tradeoff might be mitigated if, as suggested above, the memory readout could pool across multiple discretized representations with different locations of staircase steps so that the summed output had finer steps. Our work complements traditional attractor models of working memory that typically fall into two classes: bistable models that robustly maintain two levels of activity (Figs. 1b and 2b) and continuous attractor models that can maintain nearly analog storage of memory but require very precise tuning of connection weights (Figs. 1d, e and 5c, d). Our model represents an intermediate possibility with relatively moderate tuning requirements (Fig. 5b) and a discretely graded set of response levels. Previous work has suggested how multiple, spatially distinct bistable processes in a cell can be coupled together to form multiple stable levels of firing activity[45,46]. Here, we demonstrate a complementary mechanism for forming multistable representations that relies on temporal, rather than spatial, patterning of inputs. Altogether, this work suggests that the oscillatory activity commonly observed during working memory tasks may expand short-term memory capacity by enabling multi-level storage of information in persistent or sequential activity.

## Methods

The Wang-Buzsaki (W-B) model neuron used for most spiking neuron simulations is based upon the original model described in ref. [33]. Below, we show the equations for the dynamical variables most relevant to the maintenance of discretely graded persistent activity. The full model equations, and modifications of the equations below to include input noise, stochasticity in the parameters, or more complex forms of oscillatory inputs with heterogeneous phases or time-varying amplitudes and frequencies across neurons, are included in the Supplementary Information. The membrane potential of the Wang-Buzsaki neuron obeys the current balance equation:

$$C_m \frac{dV_i}{dt} = -I_{Na}(V_i, h_i) - I_K(V_i, n_i) - I_L(V_i) + I_{syn,i}(s_1, \ldots, s_N) + I_0 + I(t) + I_{ext,i}(t) \tag{1}$$

$$I_{syn,i}(s_1, \ldots, s_N) = \sum_{j=1}^{N} w_{ij} s_j \tag{2}$$

$$\tau_{syn} \frac{ds_i}{dt} = -s_i + \alpha_{syn} \sum_{t_i^{spike}} \delta\left(t - t_i^{spike}\right) \tag{3}$$

$$I(t) = \psi \cos(\omega t) \tag{4}$$

where $h$ and $n$ are time-varying channel variables (Supplementary Note 1). The parameter values used are specified in Supplementary Tables 1–3. The Wang-Buzsaki neuron receives several sources of inputs: (1) $I_{syn,i}(s_1, \ldots, s_N)$ represents recurrent feedback to neuron $i$, the strength of which is determined by a weight matrix $w_{ij}$ defining the strength of the connection from neuron $j$ to neuron $i$, (2) $I_0$ is a constant current that shifts the resting potential, and could represent tonic background input or intrinsic currents not explicitly modeled, (3) $I(t)$ is the external oscillatory input ($I(t) = 0$ for models with no oscillatory input), and (4) $I_{ext,i}(t)$ represents the external inputs to be accumulated and stored by the memory network. To calculate spike times in Eq. (3), we used the time of the peak of the action potential, with only action potentials exceeding a voltage of 0 mV included. Integration was performed numerically using the fourth order Runge-Kutta method with a time step $\Delta t = 10^{-2}$ ms.

In the single neuron case, there is a single recurrent synaptic weight $w[\mu A/cm^2]$. Values for simulation parameters are included in Supplementary Table 2. In Figs. 6–8, we study three different network architectures composed of Wang-Buzsaki neurons: an all-to-all connectivity (Fig. 6), a ring structure (Fig. 7), and a

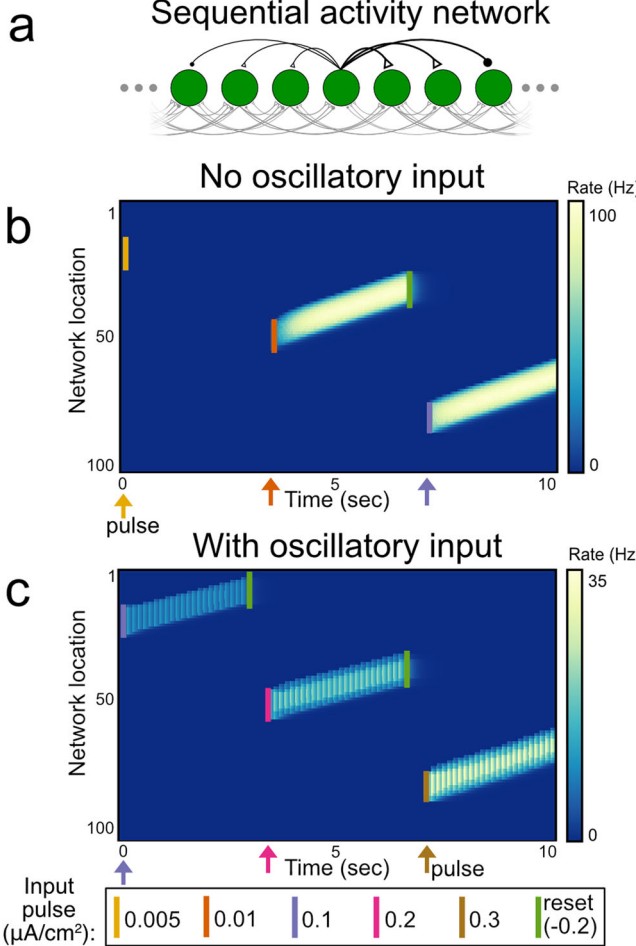

**a** Sequential activity network

**b** No oscillatory input

**c** With oscillatory input

Input
pulse
($\mu$A/cm$^2$): | 0.005 | 0.01 | 0.1 | 0.2 | 0.3 | reset (-0.2) |

**Fig. 8 Generation of sequences of discretely graded activity in a network of W-B neurons with asymmetric connectivity. a** Schematic of network structure. Asymmetric connectivity underlies slow drift of activity bumps. **b** Illustration of low-rate problem in a sequential-activity network of conductance-based spiking neurons without oscillatory input. Drifting bumps of activity in the network initiated by short (100 ms) pulses (labeled by colored bars) exhibit only a single level of activity. **c** Sequential-activity network with oscillatory input is able to maintain drifting bumps with discretely graded levels of activity.

directed structure (Fig. 8). Simulation parameter values specific to multi-neuron networks are included in Supplementary Table 3.

The all-to-all connected networks of Fig. 6 are composed of 1000 Wang-Buzsaki neurons. Figure 6b, c implements a network with uniform connection strengths $w_{ij} = \frac{w}{N}[\mu A/cm^2]$. Figure 6d implements a network in which these uniform connection strengths have been perturbed by adding static Gaussian noise of mean zero independently to each connection. Exponentially filtered temporally white noise (Ornstein-Uhlenbeck process) input was implemented in the network illustrated in Fig. 6c; for each neuron $i$, the additive noise was given independently by:

$$n_i(t) = n_{i,t-\Delta t} - n_{i,t-\Delta t}\frac{\Delta t}{\tau_n} + \sigma_n \eta_{i,t}\sqrt{\frac{\Delta t}{\tau_n}} \quad (5)$$

$$\eta_{i,t} \sim N(0,1) \quad (6)$$

where $\sigma_n$ is the amplitude of the noise.

For the ring connectivity structure in Fig. 7, the connection strength from neuron $j$ to neuron $i$ is described by:

$$w_{ij} = A + B\cos\left(\frac{2\pi(i-j)}{N}\right)[\mu A/cm^2] \quad (7)$$

The directed structure illustrated in Fig. 8 resembles the ring structure, but results in a drift of the "activity bump" in one direction. The connection strength

from neuron $j$ to neuron $i$ in this case is defined by:

$$w_{ij} = A + B\cos\left(\frac{4\pi(i-j)}{N} + 0.1\right)H(C - |i-j|)[\mu A/cm^2] \quad (8)$$

where $H$ is the Heaviside (step) function and $C$ controls the spatial extent of the connectivity.

**Comparison to linear spiking autapse model.** In Fig. 5, we compare the robustness of the discretely graded persistent activity of the nonlinear spiking model described above, to that of an approximately linear spiking autapse model in which analog persistent activity is enabled by excitatory feedback that is tuned to offset the intrinsic decay of activity. The equations for the linear spiking autapse model[36] are included in Supplementary Note 1, with parameter values in Supplementary Table 4.

**Simple rate model.** The equation for the simple firing rate model implemented for Fig. 1 is given in Fig. 1a. The nonlinear term used for Fig. 1b is:

$$f(x) = \sqrt{w[x - x_{thr}]_+} \quad (9)$$

with $w = 75$, $x_{thr} = 10$[Hz].

**Integrate and fire model.** In Fig. 4, we use an integrate-and-fire model whose membrane potential ($V$) dynamics are given by:

$$\frac{dV}{dt} = -\frac{1}{\tau}(V - V_L) + I_{syn}(s) + I_0 + I(t) + I_{ext}(t) \quad (10)$$

where $\tau = 10$ ms is the time constant of the leak when present, $I(t)$ is the oscillatory input (given by Eq. (4)) with amplitude $\psi = -1.0$ $\mu$A/cm$^2$ and $\omega = 0.05$ ms$^{-1}$, $I_{syn}(s)$ (Eq. (2)) is the synaptic input, which was held constant (parameter values given in Supplementary Note 3), $I_0 = -0.4$ $\mu$A/cm$^2$ is a constant current that shifts the resting potential as in Eq. (1), and $I_{ext}(t)$ represents an external input. We used a spike threshold of $-59.9$ mV and voltage reset of $-68$ mV; we did not enforce a refractory period.

**Reporting summary.** Further information on research design is available in the Nature Portfolio Reporting Summary linked to this article.

## Data availability
Source data to generate all figures is provided on a GitHub repository at https://github.com/goldman-lab/Championetal2023.jl.git and https://doi.org/10.5281/zenodo.8170156[47].

## Code availability
Code to implement all simulations described in this manuscript is available at https://github.com/goldman-lab/Championetal2023.jl.git and https://doi.org/10.5281/zenodo.8170156[47].

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

## Acknowledgements

We thank Steve Luck for helpful comments on the manuscript, Michale Fee for helpful discussions, and the 2016 Methods in Computational Neuroscience course (supported by NIH grant R25 MH062204 and the Simons Foundation) at the Marine Biological Laboratory, where this collaboration was started. This work was supported by NIH grants R01EY027036 and U19NS104648 (M.S.G.), NSF Graduate Research Fellowship DGE-1256082 (K.P.C.), NSF grant DMS 1951099 (G.B.E.), and Swiss National Science Foundation no. 200020 184615 (O.G.).

## Author contributions

K.P.C., O.G., B.S.L., G.B.E., and M.S.G. created the computational model and wrote the paper. K.P.C., O.G., and B.S.L. performed the simulations. G.B.E. and M.S.G. supervised the project.

## Competing interests

K.P.C. declares that this work was completed outside current employment at Amazon.com, Inc. The remaining authors declare no competing interests.
