## [Peer Review File · Communications Biology]

Reviewers' comments:

Reviewer #1 (Remarks to the Author):

In the current manuscript Champion et al., investigate whether adding oscillatory responses to a working memory network can lead to more robust persistent activity absent of issues identified in previous models (all-or-nothing and fine-tuning issues). The paper was very well written and organized. I appreciated a lot the way the authors guide you through the issues and the model, this was all very clear. I still have some concerns regarding the proposed mechanism, clarity to the specifics of the interpretation of the output, and the neurobiological plausibility.

Major issues:

- 1) Two things seem intermingled in the manuscript. The possibility to store a bigger variety of items versus the possibility to store multiple items at the same time. As far as I see it the model only provides a solution for the former, but does not provide any simulation of how the model can store multiple items in parallel (like a grocery list). The simulation regarding the sequence does not provide a satisfying answer to this question as the input is also sequential (also see point 8 regarding the sequence). Do the authors have any suggestion how this model could implement multi-item storage? Is it rather related to different nodes of the model (spatial code) or to a temporal code (e.g. proposed by Lisman)?
- 2) Also related to multi-item storage. In the simulations the authors show that dependent on the output rate, a different item can be stored (related to the strength of the input). This means that a single node in the network cannot be active in relation to different working memory representations. It is likely that single neurons are active during the representation of multiple items, should the brain decode the information from these mixed signals or is this already separated at the representation side itself?
- 3) The whole manuscript hinges on the notion that rate represents information (as that is their output variable). Is there strong evidence that the content of a representation is related to a non-linear variation of rate? For gamma frequency there is some evidence (Roberts et al., 2013; Hadjipapas et al., 2015), although those seem rather continuous instead of multi-level. Could the authors provide literature that suggest that a rate code (and specifically the frequency of the firing) is realistic and there are clear 'modes' of firing rate frequencies (rather than an continuous change).
- 4) Related to point 3. The rate of firing seems to be always a multiple of the oscillatory input (see Figure 2). How realistic is this? Is there any neuronal evidence for this? Lakatos et al., provided some hierarchical account in the auditory cortex, but in the memory system I am not really aware of these types of harmonical hierarchies in firing rates. Also depending on the base rate of the oscillation the 'code' covers frequencies ranging from 8-80 Hz. Most oscillatory account pose very different function to these different bands. Here, they are treated as being equal.
- 5) Related point 3. There has been quite some evidence that the phase at which individual neurons fire is relevant for the coding itself (e.g. Panzeri et al., 2001; 2010). Would the model account for this? In most of the simulation, there is not too much variation in when the individual neurons fire, mostly on the network as a whole. Or at least it is not clear how the variance in the phase of firing relates to the representation itself.
- 6) In the model the external input and the oscillation are two fully independent inputs. Is this realistic or is it more likely that the oscillation itself is an intrinsic feature and consequence of the local population activity? If the oscillation is a separate rather than intrinsic property where is it coming from? If it is intrinsic then you should account for how it is influenced by the activity in the population (e.g. Winfree, 1980; Doelling & Assaneo, 2021).
- 7) All neurons have the same base firing rate. Is this realistic? I believe that the variation of the base firing rate could also account for some of the effects found in Figure S1. If some nodes already have a high base firing rate and low threshold for activation, they phase-of-firing will be different than for other nodes. This, together with interconnected network, could also account how the information is maintained across oscillatory cycles. This would at the same time create a phase-of-fire code (point 4).
- 8) In figure 6 they suggest their model can account for sequential replay of activation of information. However, in their simulation can they really provide evidence for this sequence? In my understanding they manually put in the sequence through external input. It is no wonder that the model will also be active according to the sequence. I don't think we can call this sequence

representation as found in ongoing activity in the brain. The main issue there is to account for how the brain knows how to play this sequence, and they avoid this problem by putting in the external input. While they show an interesting sweep of activation (see also Lubenov and Siapas, 2009; Buzsaki and Tingley, 2018), they don't show anything related to sequence representation.

9) In both figure 6 and 7 they impose an external reset. This is needed to stop the persistent activity. What triggers this reset? Do the authors have an idea about this. I am fine if the model does not explain or account how the persistent activity stops, but I would be explicit that this cannot be explained.

Minor issues:

10) It took me quite long to understand what is meant with multi-level storage capabilities. It would be good to first introduce this before using this term. Just to be sure from my side, you refer to the possibility of the model to reach a distinct amount of neural states (instead of a binary or analog one)?

11) How is rate calculated in Fig 2D. Rate is not constant as there is no activity during the trough of the oscillation. But even in the trough of the oscillation in figure 2D-F there is a rate. Rate always need some window to calculate, so what was this window, and is in the left of these figures only average rate represented? Being more specific on this would be helpful.

12) Both in figure S1 and S3A it is investigated how varying phase consistency across nodes influences the model. But the output figures seem to be different. Specifically, the cells are active for a more wider time range in S3A. Is this related to the shorter constant or what is the difference?

13) Would be good to also mention in the main text which model is used for which figure. Sometimes I needed to go to the supplementary information to find this info.

14) Fig S1. Isn't it more realistic that due to differential baseline firing rates you find the effect rather than randomizing the phase distribution?

15) Is it realistic that the noise for all neurons are independent? How robust is the model to correlated noise (which is likely more realistic)?

Reviewer #2 (Remarks to the Author):

Great paper that is sorely needed in helping move us along, beyond the decades-long rule of the strict rate-code hegemonists. Well I guess this paper is still proposing a rate-, not time-, code but with oscillatory overlay. Perhaps could name this type of representation -- would it be "phased rate-code" ; "interval rate-code" ??

My major concerns are minor and my minor concerns are trivial.

Major

Real networks are not fully-connected and are not chains. How well does this work with sparsely connected networks?

I appreciate that this is a theory paper but I would like to see more discussion to address the implications for understanding nervous systems - perhaps such is intended for a later more-general review paper?? The macaques generally measure rates over long periods; what periods should they be measuring? Can any of the macaque (or cat or rat or mouse) public data be reanalyzed to look for the different patterns predicted by the phased hypothesis? Perhaps reference could be made to major physiological osc-a-lots: Fries, Schroeder, etc. and how to interpret their measures.

Please provide code on modeldb (with password) so can be reviewed for clarity (ideally having some commenting and documentation).

Moderate

Fig 5.

A is unnecessary -- most people know what all-to-all means and those who don't will just be confused by this schematic. Instead of A would suggest showing some illustrative cell spikings over input osc and input step for couple cycles before and after a transition.

C (no noise) should come before noise is added.

B,D -- it would be instructive to see the breakdown after the level of max. tolerance.

btw, R side paren in '(max. tolerance)' is off of the page.

Minor

Fig 2: "External input" might be labeled "signal" or "DC signal" or "stepped signal"; "Oscillatory input" -> "Oscillatory background"

WB or W-B

Abstract -- I didn't really like it As above, I would prefer to see some more reference to neurobiology and to the contrasting implications for our understanding of the brain. OK, that's probably not practical -- I see that it's limited to 150 words. Could save some space in the basics perhaps, e.g. "Non-oscillatory memory storage is limited to either being all-or-none, or requires biologically implausible tuning precision. Here, we demonstrate that oscillatory inputs permit graded outputs without requiring precise tuning. We illustrate these advantages for all-to-all, ring and chain networks." ...

Another bigger-picture question would be the potential for utilizing a phase code within the bursts. Again this is likely to be out-of-scope for this paper.

Reviewer #3 (Remarks to the Author):

In the present manuscript the authors propose that oscillatory inputs can enable individual neurons to achieve graded (multi-level) persistent activity. This is achieved via a simple and general mechanism where neuronal action potentials phase lock to the oscillatory input and neurons can fire different number of spikes in each oscillatory period. The authors illustrate that their model, when embedded in a recurrent network is tolerant to various types of noise, and is able to maintain stable or propagating bumps of variable amplitude in a ring-network.

The paper is well written: the logic and the presentation are clear and the figures properly illustrate the concepts and the results. My main concern is that the results of the paper seem "illustrative": the authors use computer simulations to demonstrate that their model is able to generate a certain property (multi-level persistent activity). However, they do not provide a rigorous analysis regarding the necessary criteria for achieving such activity. In page 6 they list 3 prerequisites required for their model:

1. Oscillatory input must be strong to reset the activity. What this reset exactly means and how one can see whether a particular reset is strong enough is not specified.

2. Activity should be carried through. Since the neuronal activity is reset during the subthreshold part of the cycle, information should be represented somewhere else, that is not 'reset'.

3. Nonlinearity is crucial for robustness. What is the critical form of nonlinearity is not specified.

Note, that the modified Seung model is also nonlinear as it has a firing threshold around 1 uA/cm² input current. What seems to be critical here is that the oscillatory input should be strong enough restart the integration, but how this property is related to the nonlinearity of the F-I curve is not clear.

The paper would benefit from a rigorous analysis using a more abstract spiking neuron model (exponential integrate and fire) where these properties can be directly specified and analysed either numerically or analytically. I acknowledge that this would be a major extension and is probably beyond the scope of the present paper. However, more specific definition of the 'reset' and the 'appropriate form of the nonlinearity' would be highly desirable.

Specific comments:

- Fig 4A and C: representation of the input was not clear to me. The scalebar indicating the presence of the external input could be mistaken for a minus sign. The magnitude of the inputs is not clear. Same applies for Fig 5B C and D.
- Fig 4A the -5% tuning has a profound effect for small inputs. The orange trace decayed back after the first pulse to its baseline. If input at 1 and 2 s had similar magnitude, then why the second pulse had a larger effect?
- Fig 4C-D id there any reason for not showing the linear network with oscillatory inputs? - Fig S2?
- I expected that input noise in the network would decrease the difference between the discrete activity levels but would not induce systematic biases in the activity. Fig 5C shows however, that input noise always cause a systematic downward in the activity levels. Is there an intuition behind this?
- In Fig 5 C and D network activity is shown at max noise tolerance level. How this noise tolerance threshold was determined? To my eyes, the degradation is already noticeable.

Reviewer #1 (Remarks to the Author):

In the current manuscript Champion et al., investigate whether adding oscillatory responses to a working memory network can lead to more robust persistent activity absent of issues identified in previous models (all-or-nothing and fine-tuning issues). The paper was very well written and organized. I appreciated a lot the way the authors guide you through the issues and the model, this was all very clear. I still have some concerns regarding the proposed mechanism, clarity to the specifics of the interpretation of the output, and the neurobiological plausibility.

Major issues:

1) Two things seem intermingled in the manuscript. The possibility to store a bigger variety of items versus the possibility to store multiple items at the same time. As far as I see it the model only provides a solution for the former, but does not provide any simulation of how the model can store multiple items in parallel (like a grocery list). The simulation regarding the sequence does not provide a satisfying answer to this question as the input is also sequential (also see point 8 regarding the sequence). Do the authors have any suggestion how this model could implement multi-item storage? Is it rather related to different nodes of the model (spatial code) or to a temporal code (e.g. proposed by Lisman)?

We thank the reviewer for this question and the opportunity to improve our presentation of the material. First and foremost, we realized that our figure on the sequence (old Figure 7, now Figure 8) was confusing due to an unfortunate use of two very similar colors, as it did *not* have sequential input. Rather, the input triggering each sequence was a single brief pulse, and the output was then a sequence of neuronal responses. Thus, the model can generate sequences that could represent multiple items. We have now fixed the color scheme as shown in the figure below. Also note in this figure that, within a single simulation, we illustrate the network generating 3 different, independent sequences of varying magnitude (one initiated by a small pulse at time 0, one initiated by a medium-amplitude pulse at time 3.5 seconds, and one initiated by a larger-amplitude pulse at time 7 seconds):

More generally, throughout the manuscript, we focused on a graded representation of one memory at a time; however, narrower lateral inhibition (for example, in the ring network presented in Fig 6, now Figure 7) can easily permit more memories to be stored by a spatial code with multiple ‘bumps’ of varying amplitude, as well as actively removed from memory (‘resets’ in figure below) without disrupting other currently stored memories:

2) Also related to multi-item storage. In the simulations the authors show that dependent on the output rate, a different item can be stored (related to the strength of the input). This means that a single node in the network cannot be active in relation to different working memory representations. It is likely that single neurons are active during the representation of multiple items, should the brain decode the information from these mixed signals or is this already separated at the representation side itself?

While we have focused in this manuscript on canonical models with simple architectures, there is still an aspect of such population coding that can be seen in, for example, the ring model (Fig 6, now Fig 7). Here, amplitude and location are ‘multiplexed’ at the level of single neurons— that is, while a given single neuron ‘tuned to’ 60 degrees on the ring will respond most when the network receives an input at 60 degrees, it will also respond when the network receives any input within many tens of degrees (i.e., whatever the width of the population response is) of its optimal value. This is a simple, topographic example of more general, non-topographic (or not obviously topographic) population codes in which neurons respond at different firing rates to different stimuli.

3) The whole manuscript hinges on the notion that rate represents information (as that is their output variable). Is there strong evidence that the content of a representation is related to a non-linear variation of rate? For gamma frequency there is some evidence (Roberts et al., 2013; Hadjipapas et al., 2015), although those seem rather continuous instead of multi-level. Could the authors provide literature that suggest that a rate code (and specifically the frequency of the firing) is realistic and there are clear 'modes' of firing rate frequencies (rather than an continuous change).

Thanks for this excellent point, as we believe this is a core question at the heart of neural coding. In the present model, we choose to focus on the concept of a rate code, both for its simplicity in illustrating principles and since it has been widely suggested to represent information stored in working memory. Most relevant to the present manuscript are the working memory experiments of touch-frequency discrimination (Romo et al., 1999, 2002) and spatial location (Funahashi et al., 1989) in which working memory representations of frequency (Romo task) or spatial location (Funahashi/Goldman-Rakic task) appear to be represented in the firing rates of single units.

Decidedly less clear is whether there is evidence for discreteness of maintained firing rates in the neurons that directly generate the persistent activity underlying working memory representations. We consider this to be a falsifiable prediction of our model in any given system. Unfortunately, we are not aware of any neural recording studies of working memory that use continuous-amplitude (rather than discretely spaced) stimuli and that have localized the neurons generating the persistent neural activity. Furthermore, the identification of discretized firing rate responses may be challenging due to the presence of noise and non-stationarity in neural recordings.

We now address these points specifically in lines 297-311 in the main manuscript:

"A key experimental prediction of the model is that there should be phase-locking of neural activity to oscillations, with the number of spikes per oscillation cycle increasing in a discretized manner. To test this prediction, population-wide cellular-resolution recordings in a parametric working memory task are likely necessary given noise in single-cell activity (e.g., Fig. 6C), as well as nonstationarities. EEG recordings have identified parametric variation in power at certain frequencies that encodes the graded, parametrically varied amplitude of a stimulus [44]. However, the resolution of such recordings is insufficient to determine whether representations are discretized. A larger challenge for neural recording studies of analog working memory tasks is that the set of neurons that generate such working memory, as opposed to reading it out, has not been identified in classic graded working memory tasks [45] – a readout mechanism could nonlinearly transform the representation, or pool across multiple discretized representations to obtain higher resolution readouts. The advent of high-resolution, population-wide recordings will hopefully make possible future experimental studies that can test the theoretical principle proposed in this work."

4) Related to point 3. The rate of firing seems to be always a multiple of the oscillatory input (see Figure 2). How realistic is this? Is there any neuronal evidence for this? Lakatos et al., provided some hierarchical account in the auditory cortex, but in the memory system I am not really aware of these types of harmonical hierarchies in firing rates. Also depending on the base rate of the oscillation the 'code' covers frequencies ranging from 8-80 Hz. Most oscillatory

account pose very different function to these different bands. Here, they are treated as being equal.

With regards to the rate of neuronal firing and neuronal evidence for this, we are not aware of any published data that confirm or falsify our model but we note that the introduction of noise makes at least the individual neuron firing rates variable (while of course still modulated by the oscillatory input) as shown in our updated Figure 6C (formerly Figure 5C):

With regards to the function of these different frequency bands, we tried to remain as agnostic as possible in this paper; what we hope to convey is that the phase-locking phenomenon we describe in the paper could operate in most of the frequency bands that have been associated with working memory tasks, and illustrate the considerable difference in the number of distinct firing rates that can be achieved in different frequency bands, as well as how the associated robustness of those rates changes with frequency.

5) Related point 3. There has been quite some evidence that the phase at which individual neurons fire is relevant for the coding itself (e.g. Panzeri et al., 2001; 2010). Would the model account for this? In most of the simulation, there is not too much variation in when the individual neurons fire, mostly on the network as a whole. Or at least it is not clear how the variance in the phase of firing relates to the representation itself.

Thanks for the question. While our model was not designed to specifically encode information in the neuronal phase of firing, we do note that it can generate responses with different phases of neuronal response on different neurons. In the simple examples in the main text, our model fires bursts of spikes that are approximately centered on the peaks of the oscillatory drive. However, this oscillatory drive might have different phases across neurons (see Figure S3A) for two reasons. First, there could be scatter or gradients of phase across the oscillatory inputs to neurons. Second, the phases could differ if they are intrinsically generated within neurons and, as in our simulations of Figure S3A, sum to create an approximately uniform set of inputs to other neurons (see the essentially flat summed firing rate in Fig. S3A, top panel) so that the intrinsic oscillation phases do not phase lock over the relevant timescales. We separately note that, if the oscillation frequency is independently noisy to each neuron, this

can manifest as a shifting of the phase of different neuronal responses within each cycle (Figure S3B).

6) In the model the external input and the oscillation are two fully independent inputs. Is this realistic or is it more likely that the oscillation itself is an intrinsic feature and consequence of the local population activity? If the oscillation is a separate rather than intrinsic property where is it coming from? If it is intrinsic then you should account for how it is influenced by the activity in the population (e.g. Winfree, 1980; Doelling & Assaneo, 2021).

Because the model specifies a mechanism and does not specify a particular brain region or circuit, there are many different oscillatory sources that are compatible with our model, depending on the specific region hypothesized to generate persistent firing. The interpretation of the model is most straightforward as a population of excitatory neurons that receive a functionally feedforward oscillatory signal (in addition to their recurrent inputs), as has been suggested to functionally describe medial septal region pacemaker neurons input to the hippocampus (Hangya et al., 2009) and the multipolar bursting interneurons circuit in neocortex, which generates synchronous and rhythmic theta activity in response to cholinergic drive and is thought to provide functionally feedforward input to pyramidal neurons (Blatow et al., 2003). Alternatively, as noted by the reviewer and addressed in the response above to Reviewer point 5, the inputs could be intrinsically generated but have random phases so that the net recurrent synaptic drive from the population is approximately uniform as in Figure S3A (top).

7) All neurons have the same base firing rate. Is this realistic? I believe that the variation of the base firing rate could also account for some of the effects found in Figure S1. If some nodes already have a high base firing rate and low threshold for activation, they phase-of-firing will be different than for other nodes. This, together with interconnected network, could also account how the information is maintained across oscillatory cycles. This would at the same time create a phase-of-fire code (point 4).

We thank the reviewer for this thought-provoking suggestion. With regards to having the same base firing rate for all neurons, this is indeed a model simplification that could be relaxed. Since neurons only see the sum total drive from their neighbors, then a neuron could, for example, receive 3 spikes per cycle input from all neurons or receive an equal mix of 2, 3, and 4 spikes per cycle. To do this, would require that those firing at 2 or 4 spikes per cycle have the appropriate negative or positive tonic background input, respectively, so that they maintain their lower or higher firing rates (and potentially some neurons with positive tonic background input would have spontaneous firing rates in the absence of input). We note that our Figure 6D (old Figure 5D), although created by (frozen) noise in weights rather than background input, shows a case where neurons have different firing rates.

With regards to the phase, the above-described case would not change the feature (observed in Figure 6D, for example) that, if the oscillation is identical to all neurons, the firing would still center on this oscillation for all neurons. Alternatively, one could put a very small background input into a neuron that would shift the phase without adding an additional spike per cycle – however, the phase shifts created by this would be limited, as the firing is already limited to the highest portion of the oscillatory cycle and, if the input was too large, it would add an

additional spike (fitting with the tendency of the spiking to approximately center around the peak of the oscillation). Thus, we do not think it could create the very broad distribution of phases shown in Figure S1B,C that are more easily created by simply having the external (or potentially, intrinsic) oscillation have a phase shift. In addition, it would shift the tuning for such neurons and make them more subject to noise.

8) In Figure 6 (*note: now Figure 7*) they suggest their model can account for sequential replay of activation of information. However, in their simulation can they really provide evidence for this sequence? In my understanding they manually put in the sequence through external input. It is no wonder that the model will also be active according to the sequence. I don't think we can call this sequence representation as found in ongoing activity in the brain. The main issue there is to account for how the brain knows how to play this sequence, and they avoid this problem by putting in the external input. While they show an interesting sweep of activation (see also Lubenov and Siapas, 2009; Buzsaki and Tingley, 2018), they don't show anything related to sequence representation.

As addressed in response to Major comment #1 above, our figure colors and layout were confusing. The sequence was generated by the network dynamics, rather than being put into the network through external input.

9) In both Figures 6 and 7 (*note: now Figures 7 and 8*) they impose an external reset. This is needed to stop the persistent activity. What triggers this reset? Do the authors have an idea about this. I am fine if the model does not explain or account how the persistent activity stops, but I would be explicit that this cannot be explained.

The oscillation-enabled persistent activity could be terminated in two physiologically plausible ways. In one scenario, the oscillatory input could act as a gate that turns on and off persistent activity, so that (as we have found in our not-shown model simulations) simply removing the oscillation would cause a cessation of the persistent activity. In the other scenario, which we have used in this paper, an inhibitory input drives the persistent activity down to zero. Recent work has suggested that a 'termination signal' originating in the superior colliculus may be responsible for terminating neural integration during some tasks (Stine et al., 2022); known connectivity of the tecto-thalamo-cortical system (Benavidez et al., 2021) would suggest that this takes place via feedforward thalamo-cortical inhibitory projections (excitatory thalamic projections that preferentially target cortical PV interneurons) (Delevich et al., 2015), essentially acting as a pulse of inhibitory input.

We have now added to the caption of Figure 7:

"For simplicity, the persistent activity in panels B and C is terminated using a pulse of inhibitory input; we do not model the origin of such termination signals."

Minor issues:

10) It took me quite long to understand what is meant with multi-level storage capabilities. It would be good to first introduce this before using this term. Just to be sure from my side, you

refer to the possibility of the model to reach a distinct amount of neural states (instead of a binary or analog one)?

Thanks for this suggestion, and you are correct about the definition. We have removed “multi-level” from the abstract, where we agree it was confusing, instead replacing it by “permits many different amplitudes of persistent activity to be stored”. We then do not introduce the phrase “multi-level” until we show the ‘staircase’ shaped plots, where there are clear ‘levels’ of activity.

11) How is rate calculated in Fig 2D. Rate is not constant as there is no activity during the trough of the oscillation. But even in the trough of the oscillation in figure 2D-F there is a rate. Rate always need some window to calculate, so what was this window, and is in the left of these figures only average rate represented? Being more specific on this would be helpful.

We thank the reviewer for the opportunity to improve our clarity. The time-varying rates depicted are exponentially-filtered spike trains, using a filter time constant equal to the model’s synaptic time constant. Expressed this way, the rate is proportional to the synaptic input variable. We have added an explanation of this to the caption of Figure 2, *“Time-varying rates (B,D,E,F, right) are computed by smoothing the spike trains using an exponential filter with time constant equal to the model’s synaptic time constant (150 ms).”*

12) Both in figure S1 and S3A (*note: these figures are now S1 and S2A*) it is investigated how varying phase consistency across nodes influences the model. But the output figures seem to be different. Specifically, the cells are active for a more wider time range in S3A. Is this related to the shorter constant or what is the difference?

Thanks for the question. Figure S1B and S2A (old Figure S3A) do have similar outputs. We think the confusion is that we have zoomed in more in S2A (0.5 sec vs. 2 sec in S1B), which makes the spikes appear to occupy a wider time range.

13) Would be good to also mention in the main text which model is used for which figure. Sometimes I needed to go to the supplementary information to find this info.

Thanks for this helpful suggestion. We have updated the main text and several figure captions to clarify which model is used in each figure.

14) Fig S1. Isn’t it more realistic that due to differential baseline firing rates you find the effect rather than randomizing the phase distribution?

Thank you for the question, which is addressed in the response to Major Comment #7 above.

15) Is it realistic that the noise for all neurons are independent? How robust is the model to correlated noise (which is likely more realistic)?

Taking the simplest and most extreme case, 100% correlated noise in the larger network will behave similarly to noise in the single-neuron model (the model depicted in Figure 2D-F). The width of the staircases in Figure 2 D-F give a rough idea of the amplitude of perfectly correlated noise that the network can tolerate (if the noise is fast, i.e., does not have prolonged temporal correlations) without jumping up/down to the next fixed point. This robustness depends on the frequency of the driving oscillation, as the staircases are wider for higher frequencies. Thus, there is a tradeoff between the number of different firing rate levels the network can maintain and the robustness to noise, as we now note in Lines 316-321:

“higher frequency oscillations do not require long timescale processes to span the oscillation cycle and are more robust to noise, but due to their short period may only store one or a few values (Figure 2F); lower frequency oscillations could store more items, but require a process with longer timescale to bridge the troughs occurring in each cycle and are more easily perturbed by noise.”

We also note that the network cannot tell the difference between 100% correlated noise and an external input that it is supposed to integrate, so correlated noise of sufficient amplitude will cause the model to diffuse over time. The figure below illustrates this extreme, 100% correlated noise case for an external oscillatory input of frequency 24 Hz (the model shown in Figure 2E):

Reviewer #2 (Remarks to the Author):

Great paper that is sorely needed in helping move us along, beyond the decades-long rule of the strict rate-code hegemonists. Well I guess this paper is still proposing a rate-, not time-, code but with oscillatory overlay. Perhaps could name this type of representation -- would it be "phased rate-code" ; "interval rate-code" ??

My major concerns are minor and my minor concerns are trivial.

Major

Real networks are not fully-connected and are not chains. How well does this work with sparsely connected networks?

Thanks for bringing up this point. We show below that the mechanism still works in sparsely connected networks, as long as there are enough neurons to enable sufficient averaging of the noisy inputs created by the sparse connectivity. This is because the network's 'staircase-like' tuning condition is robust to noise, including weight noise (cf. Figure 6D, formerly Figure 5D), as long as the noise is not so large that it kicks the network from one 'step' (fixed point) of the staircase of persistently maintainable firing rates to the next.

The figure below is now included as Supplementary Figure S2.

I appreciate that this is a theory paper but I would like to see more discussion to address the implications for understanding nervous systems - perhaps such is intended for a later more-general review paper?? The macaques generally measure rates over long periods; what periods should they be measuring? Can any of the macaque (or cat or rat or mouse) public data be reanalyzed to look for the different patterns predicted by the phased hypothesis? Perhaps

reference could be made to major physiological osc-a-lots: Fries, Schroeder, etc. and how to interpret their measures.

This is an excellent question, and as the Reviewer suggests, one that we have deferred in order to get out the theoretical idea in a general fashion rather than tying it to a particular data set or experiment. With regard to the specific predictions of our model, we believe that most of the data sets currently available may be insufficient to test our hypotheses. We now address this point specifically in lines 297-311 of the main manuscript:

“A key experimental prediction of the model is that there should be phase-locking of neural activity to oscillations, with the number of spikes per oscillation cycle increasing in a discretized manner. To test this prediction, population-wide cellular-resolution recordings in a parametric working memory task are likely necessary given noise in single-cell activity (e.g., Fig. 6C), as well as nonstationarities. EEG recordings have identified parametric variation in power at certain frequencies that encodes the graded, parametrically varied amplitude of a stimulus [44]. However, the resolution of such recordings is insufficient to determine whether representations are discretized. A larger challenge for neural recording studies of analog working memory tasks is that the set of neurons that generate such working memory, as opposed to reading it out, has not been identified in classic graded working memory tasks [45] – a readout mechanism could nonlinearly transform the representation, or pool across multiple discretized representations to obtain higher resolution readouts. The advent of high-resolution, population-wide recordings will hopefully make possible future experimental studies that can test the theoretical principle proposed in this work.”

Second, in the case that working memory representations are stored through sequential activity, the challenge of disentangling discrete graded memory storage from noise becomes more severe, as the time during which a given unit fires is more limited and the handoff of the memory from one set of neurons to another makes recording a large fraction of the relevant neurons even more difficult.

Please provide code on modeldb (with password) so can be reviewed for clarity (ideally having some commenting and documentation).

We have provided our code on our lab github repository and placed a copy of the software package on modeldb.

Moderate

Fig 5. (*note: now Figure 6*)

A is unnecessary -- most people know what all-to-all means and those who don't will just be confused by this schematic. Instead of A would suggest showing some illustrative cell spikings over input osc and input step for couple cycles before and after a transition.

We thank the reviewer for this comment. We do prefer to keep some schematic of all-to-all network structure in the paper for ease of reading by the general readership, but we agree that the figure was not particularly clear, so we have updated the figure with a less confusing depiction of all-to-all connectivity:

C (no noise) should come before noise is added.

Thanks—we had the no noise case as “B”, but had awkwardly/unconventionally organized the figure to read down the left column first. We have now changed the arrangement as suggested.

B,D -- it would be instructive to see the breakdown after the level of max. tolerance.

btw, R side paren in '(max. tolerance)' is off of the page.

The 1000-neuron network that we show in Figure 6 (old Figure 5) can handle a very large amount of input noise without a sharp breakdown in behavior. We think ‘max tolerance’ was not a good choice of label for the figure, as we illustrated a case in which the noise was strong enough to cause some neurons in the network to jump between fixed points. We therefore have removed this confusing label. We also have doubled the level of noise in Figure 6C to a more extreme amount so that the network is degraded more (Figure 6D was already very extreme, indicating the strong robustness to weight noise). Conceptually, because of the shape of the staircase, the highest firing rates tend to de-tune (i.e., drift) before the lower firing rates.

Minor

Fig 2: "External input" might be labeled "signal" or "DC signal" or "stepped signal"; "Oscillatory input" -> "Oscillatory background"

Thanks for the comment. We think that keeping all three network-related currents (background, oscillatory, and synaptic) labeled as 'inputs' makes the labeling more easy to understand for a general reader. We hope this is ok.

WB or W-B

Thanks. W-B.

Abstract -- I didn't really like it As above, I would prefer to see some more reference to neurobiology and to the contrasting implications for our understanding of the brain. OK, that's probably not practical -- I see that it's limited to 150 words. Could save some space in the basics perhaps, e.g. "Non-oscillatory memory storage is limited to either being all-or-none, or requires biologically implausible tuning precision. Here, we demonstrate that oscillatory inputs permit graded outputs without requiring precise tuning. We illustrate these advantages for all-to-all, ring and chain networks." ...

Thanks for this comment. We did make a small edit to the abstract for clarity, but given that our primary goal in this paper is to clearly demonstrate the theoretical principles (see reply above), and given the 150 word limit, we kept the abstract focused on the theoretical principle.

Another bigger-picture question would be the potential for utilizing a phase code within the bursts. Again this is likely to be out-of-scope for this paper.

Thanks for the question-- we think that this would be an exciting direction for future work. While our model was not designed to specifically encode information in the neuronal phase of firing,

we do note that it can generate responses with different phases of neuronal response on different neurons, potentially permitting phase-coding. In the simple examples in the main text, our model fires bursts of spikes that are approximately centered on the peaks of the oscillatory drive. However, this oscillatory drive might have different phases across neurons (see Figure S2A, formerly Figure S3A) due to either gradients of phase across the oscillatory inputs to neurons or due to intrinsic oscillations with different phases. This could potentially serve as a substrate for a phase code.

Reviewer #3 (Remarks to the Author):

In the present manuscript the authors propose that oscillatory inputs can enable individual neurons to achieve graded (multi-level) persistent activity. This is achieved via a simple and general mechanism where neuronal action potentials phase lock to the oscillatory input and neurons can fire different number of spikes in each oscillatory period. The authors illustrate that their model, when embedded in a recurrent network is tolerant to various types of noise, and is able to maintain stable or propagating bumps of variable amplitude in a ring-network.

The paper is well written: the logic and the presentation are clear and the figures properly illustrate the concepts and the results. My main concern is that the results of the paper seem "illustrative": the authors use computer simulations to demonstrate that their model is able to generate a certain property (multi-level persistent activity). However, they do not provide a rigorous analysis regarding the necessary criteria for achieving such activity. In page 6 they list 3 prerequisites required for their model:

1. Oscillatory input must be strong to reset the activity. What this reset exactly means and how one can see whether a particular reset is strong enough is not specified.
 2. Activity should be carried through. Since the neuronal activity is reset during the subthreshold part of the cycle, information should be represented somewhere else, that is not 'reset'.
 3. Nonlinearity is crucial for robustness. What is the critical form of nonlinearity is not specified.
- Note, that the modified Seung model is also nonlinear as it has a firing threshold around 1 $\mu\text{A}/\text{cm}^2$ input current. What seems to be critical here is that the oscillatory input should be strong enough restart the integration, but how this property is related to the nonlinearity of the F-I curve is not clear.

The paper would benefit from a rigorous analysis using a more abstract spiking neuron model (exponential integrate and fire) where these properties can be directly specified and analysed either numerically or analytically. I acknowledge that this would be a major extension and is probably beyond the scope of the present paper. However, more specific definition of the 'reset' and the 'appropriate form of the nonlinearity' would be highly desirable.

Thank you for pointing out our lack of clarity and for this suggestion of analyzing a simpler abstract spiking model, which we agree provides very useful insights. Based on these comments, we now have separated the set of critical criteria into three aspects, one each corresponding to the synaptic time constant, the oscillation strength, and the membrane potential nonlinearity. Furthermore, we have added a simulation with a simple leaky

integrate-and-fire neuron that provides mechanistic and analytic intuition for the phase locking. These are as follows:

- 1) *Sufficiently long synaptic time constant.* The synaptic time constant (reviewer point 2 above) is as the reviewer states: during the portion of the oscillation cycle when the neuron or network of neurons is not firing, the activity from the previous cycle is stored in the synaptic activation. If the time constant is too short, then this activation will not carry through to the next cycle. (And, as noted in the paper and shown in Supplemental Figure S1, this requirement becomes less strict if different neurons in a network have different oscillation phases, as then activity can be ‘carried through’ by handing off activity from one neuron to another that fires at a later phase).
- 2) *Sufficiently large oscillation strength.* The previous description of a ‘reset’ here was confusing. A more straightforward explanation is that, if the oscillation is not sufficiently large, then the network is stuck with two possibilities: similar to what occurs in the network with no oscillations, external input is either too weak to have firing in the absence of the external input or the input is so large that it goes to an extremely high firing rate ‘up state’. With the presence of a sufficiently large oscillation (but not so large that it causes firing on its own in the absence of recurrent synaptic input), the positive-going part of the oscillation gives a window of opportunity when the neuron can fire in response to its recurrent synaptic feedback (more quantitatively, this corresponds to being able to transiently move the dynamics above the unstable fixed point, which now shifts up and down with the external oscillatory input, since the effect of this oscillatory input is, roughly, equivalent to shifting the feedback curve of Figure 2B left and right). The negative-going part of the oscillation then brings activity back down (below the unstable fixed point) so that it does not get stuck in the extremely high firing rate upper fixed point seen in the no-oscillation case.

We have now added this to the main text (Lines 151-167):

“First, the oscillation must be sufficiently strong. For very small oscillatory input amplitudes, the system resembles the “no oscillatory input” case of Figure 2B, in which external input is either too weak to cause sustained spiking so that activity returns quickly to the lower, no-spiking stable fixed point, or is strong enough to trigger spiking but then runs off to the very high upper fixed point. To maintain discretely graded persistent activity, the oscillation must be strong enough at its peak to destabilize the no-spiking fixed point and cause spiking, and negative enough at its trough to de-stabilize the upper fixed point and terminate spiking in each cycle.”

- 3) *Intrinsic nonlinearity of membrane leak as a substrate for phase locking.* Phase locking is a property of nonlinear, but not linear oscillators. The key feature of this nonlinearity is that, across a cycle of the oscillation, it keeps small parameter changes from leading to phase relationships drifting apart across cycles. We now show that the ‘leakiness’ of the membrane, when sufficiently strong (i.e. sufficiently short membrane time constant), is sufficient to accomplish this, because it enables any differences in phase that built up across the spiking portion of the cycle to be forgotten on a time scale corresponding to the membrane time constant.

Specifically, taking up your suggestion about using an integrate-and-fire model, we consider a leaky integrate-and-fire neuron versus a non-leaky integrate-and-fire neuron (the latter having a perfectly linear, non-decaying summation of inputs in its subthreshold dynamics). In the non-leaky integrate & fire neuron (new Fig. 4A of the main text, reproduced below), a brief pulse of input (red, positive; yellow-orange, negative; blue, trace with no pulse) leads to a change in response phase that persists across subsequent cycles of the oscillation (Fig. 4B). As a result, if we now consider a fixed synaptic input (a simplification for pedagogical purposes approximating what would occur for a long synaptic time constant that averages across previous cycles), then even a small change in synaptic input can lead to an increase in firing rate (panel C, pink trace has slightly more synaptic input than the cyan trace). In detail, this occurs because, due to the slightly larger synaptic input, the pink trace's phase precesses backwards relative to the cyan trace until there is enough time for a second spike to occur in the cycle (and then, because this second spike occurred late in the cycle, only a single spike can fit in the subsequent cycle). By contrast, for the leaky integrate-and-fire neuron (panel D-F), the effect of a brief pulse of input is rapidly forgotten on the time scale of the membrane time constant. Indeed, for the subthreshold dynamics between spikes the voltage difference ΔV between the traces with and without a pulse can be shown to obey the equation:

$$\tau d(\Delta V)/dt = -\Delta V,$$

i.e. the effect of the phase difference in spiking decays away exponentially with time constant τ (panel E). As a result of this decay, small changes in synaptic input do not build up across cycles, so that the effect of an increased synaptic drive may lead simply to a shift back in phase of spiking (compare pink and cyan traces for a much larger change in synaptic input than in panel C) rather than to increased activity (panel F, right). For sufficiently large increase in synaptic input, an extra spike will be added and maintained. Altogether, this leads to the staircase-like response (panel F, left) that was similarly observed in the fully conductance-based Wang-Buzsaki model simulation.

The following changes have been made to the text to reflect the above point (Main text, lines 182-191):

“Third, the single neuron model must be sufficiently nonlinear to enable phase locking to the external oscillation (Fig. 3). The key feature of this nonlinearity is that it must keep small changes in synaptic input, for example due to small changes in synaptic weights, from causing corresponding changes in firing rate. This occurs in the Wang-Buzsaki model (Fig. 3A) because small perturbations (Fig. 3B, + or – pulse) that cause transient phase shifts during the supra-threshold portion of the oscillatory cycle quickly decay away during the sub-threshold portion of the cycle (Fig. 3B). Due to this decay, the number of spikes per cycle, and thus the output firing rate, can remain constant even when the level of sustained synaptic input changes (Fig. 3C), leading to the observed staircase in the firing rate versus feedback relation (Fig. 3A).

To understand more quantitatively the origin of this nonlinear phase-locking and the resultant staircase of firing rate versus feedback, we considered a simple integrate-and-fire model neuron. For a non-leaky integrate-and-fire neuron (Fig. 4A), which has perfectly linear, non-decaying subthreshold voltage dynamics, applying a brief pulse of external input led to a phase shift in spiking that persisted across cycles (Fig. 4B). Due to this lack of phase-resetting, when the level of sustained synaptic input is even slightly increased, it causes a progressive phase advance of spiking that builds up across cycles until an additional spike is added (Fig. 4C, right, compare pink and cyan traces), increasing the firing rate (Fig. 4C, left; note that, because the second spike occurred late in the cycle, only a single spike can fit in the next cycle, so the firing rate increases only slightly). For a leaky integrate-and-fire neuron (Fig. 4D), by contrast, a brief pulse of external input leads to a transient shift in the times of the next spikes, but this phase shift decays away exponentially with the leak time constant during the subthreshold portion of the cycle (Fig. 4E; mathematical derivation shown in Supplementary Information 1.4). As a result, small changes in synaptic input do not build up across cycles and quite different synaptic inputs can lead to the same number of spikes each cycle (Fig. 4F).”

Main text, Lines 371-378:

Integrate & fire model

In Figure 4 we use an integrate-and-fire model whose membrane potential (V) dynamics are given by:

$$\frac{dV}{dt} = -\frac{1}{\tau}(V - V_L) + I_{syn}(s) + I_0 + I(t) + I_{ext}(t) \quad (10)$$

Where $\tau=10$ ms is the time constant of the leak when present, $I(t)$ is the oscillatory input (given by Eq. 4) with amplitude $-1.0 \mu\text{A}/\text{cm}^2$ and frequency 8 Hz, $I_{syn}(s)$ (Eq. 2) is the synaptic input, $I_0 = -0.4 \mu\text{A}/\text{cm}^2$ is a constant current that shifts the resting potential as in Eq. 1, and $I_{ext}(t)$ represents an external input. We use a spike threshold of -59.9 mV and voltage reset of -68 mV; we did not enforce a refractory period.

Supplementary Information, Section 1.4:

“In Figure 4 of the main text, we show that the phase-locking and associated staircase of firing rate versus synaptic input occur in a leaky, but not a non-leaky integrate-and-fire neuron receiving oscillatory input. The key results were that: (1) phase differences caused by a transient pulse of input decay away exponentially during the subthreshold period in the leaky, but not the non-leaky, integrate-and-fire neuron, and (2) the leaky, but not the non-leaky, integrate-and-fire neuron displays the staircase of firing rate versus synaptic input wherein small differences in synaptic input on a given step of the staircase do not cause corresponding changes in firing rate (i.e., # of spikes per cycle). These properties can be seen analytically by examining the differences ΔV between the voltage responses of two integrate-and-fire neuron simulations defined by the following equations:

$$\begin{aligned}
\frac{dV_1}{dt} &= -\frac{1}{\tau}(V_1 - V_L) + I(t) + I_0 + I_{syn1} + I_{pulse}(t) \\
\frac{dV_2}{dt} &= -\frac{1}{\tau}(V_2 - V_L) + I(t) + I_0 + I_{syn2} \\
\frac{d\Delta V}{dt} &= -\frac{1}{\tau}(\Delta V) + \Delta I_{syn} + I_{pulse}(t)
\end{aligned}
\tag{S.7}$$

where $\Delta V = V_1 - V_2$ and $\Delta I_{syn} = I_{syn1} - I_{syn2}$. Here, as in Eq. 10 of the main text, $I(t)$ represents the oscillatory input, I_0 represents constant background input, and I_{syn} represents the synaptic input. In the simulations of the response to a pulse of input (Figs. 4B,E), $I_{pulse}(t)$ represents the brief pulse of positive (red trace) or negative (orange trace) input and the synaptic input is identical for each simulation ($I_{syn1} = I_{syn2}$)

For the non-leaky integrate-and-fire neuron, $\tau \rightarrow \infty$ so the first, exponential decay governing term in the voltage difference equation (Eq. S.7) is zero. As a result, for the simulations of Fig. 4B with a brief pulse of input, the voltage difference and corresponding phase shift caused by the pulse of input persists indefinitely. For the simulations of Fig. 4C, in which the two simulations have different synaptic inputs ($\Delta I_{syn} \neq 0$), this leads to a steady accumulation (i.e., temporal integration) of voltage difference between the two (pink and cyan) traces in the subthreshold dynamics, and a corresponding difference in firing rates.

For the leaky integrate-and-fire neuron, voltage differences in the subthreshold dynamics decay away with time constant τ . Thus, for the simulation of Fig. 4E with a brief pulse of input, the pulse causes only a transient change in firing that decays away exponentially during the subthreshold portion of the oscillation. As long as the exponential decay time constant is much less than the subthreshold period between spiking events, the voltage difference will decay away nearly completely (Fig. 4E). We note that, during the spiking that occurs near the peak of the external oscillation, the voltage differences also exhibit exponential decay between spikes, but the voltage reset caused by the spike then separates the voltage traces from each other and switches which neuron's voltage is higher, so that the phase differences do not steadily decay away during this period of rapid spiking. For the simulations with different synaptic inputs ($\Delta I_{syn} \neq 0$), the exponential decay dynamics lead to a steady-state offset in voltage $\Delta V \rightarrow \Delta I_{syn}$. This leads to the observed relative phase shift of the pink and cyan traces in Fig. 4F. However, as long as the voltage difference caused by the different synaptic inputs is not too large, it can phase shift the timing of spikes without permitting the addition of an extra spike; as a result, there is a constant 'plateau' in the firing rate for different levels of synaptic input. For larger synaptic input differences, an extra spike will be added, corresponding to the stepping up between plateaus of the firing rate versus synaptic input staircase."

Specific comments:

- Fig 4A and C (*note: now Figure 5A and C*): representation of the input was not clear to me. The scalebar indicating the presence of the external input could be mistaken for a minus sign. The magnitude of the inputs is not clear. Same applies for Fig 5B C and D.
- Fig 4A (*note: now Figure 5A*) the -5% tuning has a profound effect for small inputs. The orange trace decayed back after the first pulse to its baseline. If input at 1 and 2 s had similar magnitude, then why the second pulse had a larger effect?

We address these two points together. With regards to the symbol indicating the presence of external input, we have now replaced the confusing bars that looked like minus signs with (+) or (-) to denote positive and negative pulses, respectively (see corrected Figure below).

With regards to the magnitudes: For the -5% tuning, the same magnitude of pulse (due to the mistuning) just has slightly less effect, so the model wasn't hitting the next higher stable fixed point. If the pulse was a little stronger, the traces would look identical. Our larger point was that the -5% model can hold the same steady states, even if it takes slightly more input to reach those steady states from below. Since the main point of the figure is about holding the multiple levels rather than the slight difference in required pulse size to jump between fixed points, we now use slightly larger pulse sizes for the negatively tuned case and show that the persistent activity can be maintained across these fixed points, even with mistuning.

- Fig 4C-D (note: now Figure 5C,D) id there any reason for not showing the linear network with oscillatory inputs? - Fig S2 (note: Figure S2 has been removed and replaced)?

The main point of that figure was to show that the oscillatory model could hold multiple levels of persistent activity with weight de-tuning that would result in instability in a more classic (Seung autapse model) line attractor model.

A direct comparison of oscillations in the two models is difficult. The Seung autapse model requires fine tuning in order to achieve its approximately linear firing rate versus synaptic input (F-I) relationship. The Seung autapse model with oscillatory inputs that was previously illustrated in Fig S2 used a much smaller oscillatory input, in order to be able to keep the F-I curve close to linear. This makes it hard to compare to the Wang-Buzsaki simulation as the lack of phase-locking could simply be because the oscillation is too small. Unfortunately, because the Seung autapse model requires fine tuning, increasing the oscillatory strength interacts with the intrinsic biophysical parameters and, in order to have the oscillatory mechanism work in the Seung autapse model with larger oscillation strengths requires changing the Seung autapse model parameters to a regime where the Seung autapse model is also nonlinear.

Thus, overall, we realized that the Seung autapse model was not a great example for illustrating the effects of linearity versus nonlinearity. Rather, we think the leaky versus non-leaky integrate-and-fire example that you suggested is much more to the point for

explaining the key nonlinearity that makes the phase-locking work in this model, so we have removed the Seung autapse model with oscillatory input.

- I expected that input noise in the network would decrease the difference between the discrete activity levels but would not induce systematic biases in the activity. Fig 5C (*note: now Fig 6*) shows however, that input noise always cause a systematic downward in the activity levels. Is there an intuition behind this?

Good observation. The systematic bias in this instance is linked to the (mis-)centering of the staircases shown in Figure 5B (old Fig 4B) in relation to the 45-degree line depicting unity of feedback and rate decay; in the range of firing rates covered in this simulation, the staircase is slightly offset from this line, which biases the direction of drift.

- In Fig 5 C and D (*note: now Fig 6 C and D*) network activity is shown at max noise tolerance level. How this noise tolerance threshold was determined? To my eyes, the degradation is already noticeable.

Thanks for the question. The 1000-neuron network that we show in Figure 6 (old Figure 5) can handle a very large amount of input noise without a true breakdown in behavior. We think 'max tolerance' was not a good choice of label for the figure, as we illustrated a case in which the noise was strong enough to cause some neurons of the network to jump between fixed points, and had described this in the caption as "the point at which the network activity noticeably began degrading", which conflicted with the term 'maximum tolerance'. We have removed the confusing label from the figures.

REVIEWERS' COMMENTS:

Reviewer #1 (Remarks to the Author):

Thank you very much for this elaborate revisions. Overall I was very pleased with the answers.

I just have a few final suggestions.

Regarding point 1+2: I understand now that multiple items can be held in memory in parallel from their useful figure in the response letter. Then the authors respond that that a single neuron can respond to multiple items at the same time at different firing rates. I agree that this does mean that the networks store the items in parallel, however to me there has to be some notion of where this information is then read out. If there are parallel rates, you assume the brain has some FFT-like monitor somewhere. I think it would be good to write this clearly in the discussion.

Regarding point 4 (function of different frequency bands): I also think it would be fair to add this as a discussion point.

Regarding point 5/7: I understand this might be out of the scope of the current paper, but I still wonder how this all would relate to phase-of-firing theories as also reviewer 2 points out. But I think this does mean that there is still a lot of work to do.

Reviewer #2 (Remarks to the Author):

Accept

Reviewer #3 (Remarks to the Author):

In the revised version of the manuscript the authors carefully addressed all my previous comments. I particularly like the direct comparison of the leaky versus non-leaky integrate and fire neurons. I found that this paper is now ready for publication. I just have a few minor points that the authors may wish to consider to discuss.

- line 164: 'the oscillation must be high enough at its peak to annihilate the no-spiking fixed point and cause spiking,' - as far as I understand, this is only true in the presence of recurrent synaptic inputs.

- line 244: 'noisy oscillation frequency', S3B: 'Noise is independent (uncorrelated) across neurons of the network.' - Oscillations in the brain are typically not metronomic - i.e., their frequency vary in time or their period change from cycle to cycle. Importantly, these changes can be coherent for neighboring neurons - e.g., theta periods are correlated in neighboring electrodes in the hippocampus (Bullock et al., 1990). My intuition tells that the proposed mechanism of working memory is especially sensitive to correlated variations in the oscillation frequency. This issue could be discussed.

- line 307 (also Fig 5D and in response to comment of Rev. #1): 'or pool across multiple discretized representations to obtain higher resolution readouts.' I really liked this idea, that different neurons might get different weights of recurrent inputs, and can have different firing rates. In general, each neuron can have a different excitability or input weight, and thus the 'steps' in their 'staircases' might be at completely different levels of input. When averaged across a large population of neurons, that switch activity at different input levels, the activity of the whole population could accurately track the input. As a result, the amplitude of the bump, which was discretized to integer multiples of the oscillation frequency at the single neuron level, can be almost continuous at the population level - if neurons are sufficiently heterogeneous. This possibility could be discussed.

REVIEWERS' COMMENTS:

Reviewer #1 (Remarks to the Author):

Thank you very much for this elaborate revisions. Overall I was very pleased with the answers.

I just have a few final suggestions.

Regarding point 1+2: I understand now that multiple items can be held in memory in parallel from their useful figure in the response letter. Then the authors respond that that a single neuron can respond to multiple items at the same time at different firing rates. I agree that this does mean that the networks store the items in parallel, however to me there has to be some notion of where this information is then read out. If there are parallel rates, you assume the brain has some FFT-like monitor somewhere. I think it would be good to write this clearly in the discussion.

Thanks for bringing up this interesting point. As the reviewer notes, our previous reply showed how, in the context of a ring network: (1) multiple items can be stored simultaneously in the same network, and (2) that a single neuron responds (with different firing rates) to different stored items.

In the previous reply, we did not consider the combination of (1) and (2) above, namely how a single neuron responds when multiple items are simultaneously stored in working memory. This is a generally interesting question for working memory that previously has been suggested to relate to limitations on the number of items that can be simultaneously encoded in working memory. The key issue is whether interference occurs when the population patterns of activity for two (or more) items overlap. At least in the context of ring models built from classic recurrent excitation among similarly tuned items and broad lateral inhibition, such interference does occur and limits the capacity of working memory. As shown below in Figure 2 from the previously published classic ring model of Wei, Wang, and Wang (*J. Neurosci.*, 2012), the single neurons tend to primarily respond for a single item, and if too many items are presented, one of two things happen (see lower right panel below, showing attempting to store 6 items in working memory rather than the 2-4 items shown in the other panels): either nearby memories 'merge' due to recurrent excitation between the nearby memories (numbering the memories from top to bottom as stored items 1 through 6, see merger of items 2 and 3) or the lateral inhibition from the more strongly stored items lead to the less strongly stored items losing their activity (thus being forgotten) as occurs with items 1, 4, and 6.

One possible question suggested by the reviewer is whether some of this interference between memories could be lessened if the storage of different items occurred at different frequencies (with, as the reviewer suggests, readout possible through a Fourier transform-like operation). We think this is an interesting possibility, but it also feels to us beyond the scope of the current paper and something we would want to treat in a more thorough and general framework enabling the rigorous study of capacity limitations of oscillatory working memory networks; as such, we would prefer to leave this complex topic for future work.

Regarding point 4 (function of different frequency bands): I also think it would be fair to add this as a discussion point.

Thanks for this suggestion. We have now expanded our discussion to clarify this, as shown in the quote from the paper below:

(lines 467-473): “The mechanism can operate at any of the many oscillation frequencies suggested to correlate with working memory storage (Figure 2d-f): higher frequency oscillations do not require long timescale processes to span the oscillation cycle and are more robust to noise, but due to their short period may only store one or a few values; lower frequency oscillations could store more items, but require a cellular or network process with longer timescale to bridge the troughs occurring in each cycle and are more easily perturbed by noise.”

Regarding point 5/7: I understand this might be out of the scope of the current paper, but I still wonder how this all would relate to phase-of-firing theories as also reviewer 2 points out. But I think this does mean that there is still a lot of work to do.

We agree that this is an interesting question, but also agree that this gets beyond the scope of the current paper. We therefore leave this to future work that can delve more thoroughly into this question.

Reviewer #2 (Remarks to the Author):

Accept

Reviewer #3 (Remarks to the Author):

In the revised version of the manuscript the authors carefully addressed all my previous comments. I particularly like the direct comparison of the leaky versus non-leaky integrate and fire neurons. I found that this paper is now ready for publication. I just have a few minor points that the authors may wish to consider to discuss.

- line 164: 'the oscillation must be high enough at its peak to annihilate the no-spiking fixed point and cause spiking,' - as far as I understand, this is only true in the presence of recurrent synaptic inputs.

-Thanks and, yes, we did not mean that the oscillation was high enough to cause spiking on its own. Rather, we meant to cause spiking in the presence of the recurrent synaptic input. We now clarify this as follows:

(lines 223-230, change underlined:) “To maintain discretely graded persistent activity in the recurrently connected network, the oscillation must be high enough at its peak to annihilate the no-spiking fixed point and cause spiking, and low enough at its trough to annihilate the upper fixed point and terminate spiking in each cycle.

- line 244: 'noisy oscillation frequency', S3B: 'Noise is independent (uncorrelated) across neurons of the network.' - Oscillations in the brain are typically not metronomic - i.e., their frequency vary in time or their period change from cycle to cycle. Importantly, these changes can be coherent for neighboring neurons - e.g., theta periods are correlated in neighboring electrodes in the hippocampus (Bullock et al., 1990). My intuition tells that the proposed mechanism of working memory is especially sensitive to correlated variations in the oscillation frequency. This issue could be discussed.

Thanks for this comment. We agree that the effects of noise in frequency are lessened by the network being able to average across independent noise presented to each neuron. With regards to this, we think of real networks as having much more than 1000 neurons so, crudely, the 1000 neurons in our model can be thought of as a very simple noise model in which clusters of neurons with 100% correlated noise (represented by a single unit in our simulation) are then coupled to other neurons that are sufficiently far away (i.e. a correlation length scale away) to have approximately uncorrelated noise. That being said, this approximation still may be overly optimistic with regards to noise distributions that may, for example, have long tails or have much longer correlation length scales. We therefore below show the ‘worst-case scenario’ of 100% correlated oscillation-frequency noise. This is instantiated by a 1-neuron (autapse) network with the amplitude of oscillation-frequency noise the same as the oscillation-frequency noise in

Supplementary Figure 3b (equation S.47 of Supplementary Note 2). Four instantiations of this noise (from clicking and running our simulation four times in a row) are shown here. As can be seen, the drift in firing is, as expected, worse than for a 1000 neuron network with independent noise, but still not terrible for a worst-case simulation.

- line 307 (also Fig 5D and in response to comment of Rev. #1): 'or pool across multiple discretized representations to obtain higher resolution readouts.' I really liked this idea, that different neurons might get different weights of recurrent inputs, and can have different firing rates. In general, each neuron can have a different excitability or input weight, and thus the 'steps' in their 'staircases' might be at completely different levels of input. When averaged across a large population of neurons, that switch activity at different input levels, the activity of the whole population could accurately track the input. As a result, the amplitude of the bump, which was discretized to integer multiples of the oscillation frequency at the single neuron level, can be almost continuous at the population level - if neurons are sufficiently heterogeneous. This possibility could be discussed.

Thanks for this comment. With regards to combining across the discretized representations, we note that there are two places one can mix the multiple discretized representations: either within the recurrent dynamics that themselves generated the discretized representations or within the readout. If one does the mixing within the recurrent circuitry itself then, analogous to Figure 5d, one will get a feedback curve with less steps and thus a more analog-like representation; however, this recurrent implementation (as in Figure 5d) will come with reduced robustness. However, if the discretized representations are combined in a readout step, then by the mechanism delineated by the reviewer, one can both take advantage of the robustness of the discretized representations created by the recurrent circuitry and also get higher resolution representation in the final output.

We now expand upon this in the Discussion, indicating how, as described by the reviewer, the pooling across discretized representations might be able to make finer staircases

at the level of the readout and thus jointly enable robustness and a higher resolution representation in the readout:

(lines 468-475): “higher frequency oscillations do not require long timescale processes to span the oscillation cycle and are more robust to noise, but due to their short period may only store one or a few values; lower frequency oscillations could store more items, but require a cellular or network process with longer timescale to bridge the troughs occurring in each cycle and are more easily perturbed by noise. This tradeoff might be mitigated if, as suggested above, the memory readout could pool across multiple discretized representations with different locations of staircase steps so that the summed output had finer steps.”